# TEXT TO IMAGE SYNTHESIS WITH HYPER-CONDITIONAL GENERATIVE ADVERSARIAL NETWORKS

## ABSTRACT

We present HyperCGAN: a conceptually simple and general approach for text-to-image synthesis that uses hypernetworks to condition a GAN model on text. In our setting, the generator and the discriminator weights are controlled by their corresponding hypernetworks, which modulate weight parameters based on the provided text query. We explore different mechanisms to modulate the layers depending on the underlying architecture of a target network and the structure of the conditioning variable. Our method enjoys high flexibility, and we test it in two scenarios: traditional image generation (on top of StyleGAN2) and continuous image generation (on top of INR-GAN). To the best of our knowledge, our work is the first one which explores text-controllable continuous image generation. In both cases, hypernetwork-based conditioning achieves state-of-the-art performance in terms of modern text-to-image evaluation measures and human studies on COCO $256^2$ and ArtEmis $256^2$ datasets.

## 1 INTRODUCTION

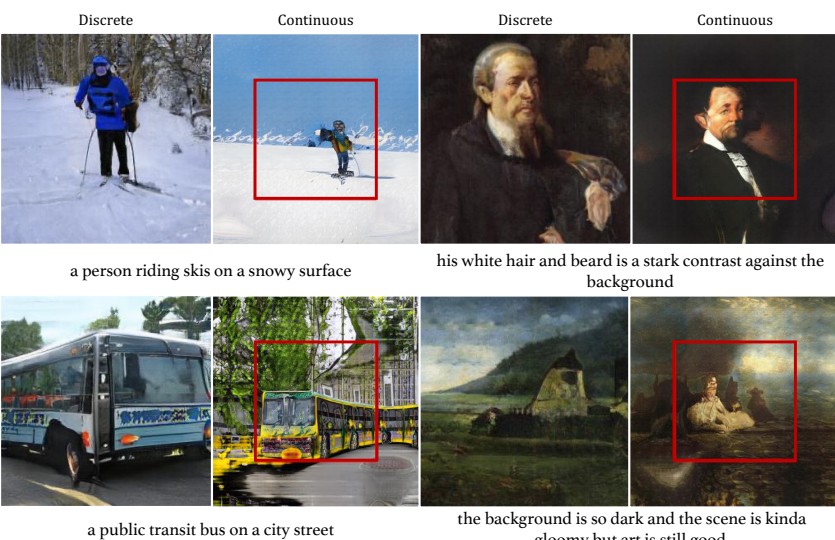

Figure 1: **Generated samples from COCO (Left) and ArtEmis (Right) using HyperNet conditioned convolution-based (discrete) and INR-based generators (continuous).** For continuous images, the regions outside the bounding boxes are extrapolations beyond the training image dimensions.

Humans have the innate ability to connect what they visualize with language or textual descriptions. Text-to-image (T2I) synthesis, an AI task inspired by this ability, aims to correctly generate an image conditioned on a textual input description. Compared to other possible inputs in the conditional

generation literature, sentences are an intuitive and flexible way to express visual content that we may want to generate. The main challenge in traditional T2I synthesis lies in learning from the unstructured description and in encoding and learning the different statistical properties of vision and language inputs. This field has seen significant progress in recent years in the quality of generated images, the size and complexity of datasets used, and in image-text alignment (Xu et al., 2018; Li et al., 2019a; Zhu et al., 2019; Tao et al., 2020; Zhang et al., 2021; Ramesh et al., 2021b).

Existing methods for T2I can be broadly categorized based on the architecture innovations developed to condition on text. Models that condition on a single caption input include stacked architectures (Zhang et al., 2017), attention mechanisms (Xu et al., 2018)], Siamese architectures (Yin et al., 2019), cycle consistency approaches (Qiao et al., 2019), and dynamic memory networks (Zhu et al., 2019). A parallel line of work (Yuan & Peng, 2019; Souza et al., 2020; Wang et al., 2020) looks at adapting unconditional models for T2I synthesis. In this work, we propose the HyperCGAN — a single-stage pipeline and general framework for T2I generation, which can efficiently generate both standard and continuous images from text. We show the generality of our method using the INR-GAN (Skorokhodov et al., 2021a) backbone for generating continuous images and the Style-GAN2 (Karras et al., 2020) backbone for generating discrete pixel-based images. StyleGAN2 is a state-of-the-art model for discrete image generation and automatically learns and separates the high-level attributes of and stochastic variations in the generated images, while INR-GAN is a recently introduced unconditional continuous image generator that significantly reduces the gap between continuous image GANs and pixel-based ones. Figure 10 shows generated discrete and continuous images on two datasets using the HyperCGAN framework.

**What is HyperCGAN?** Contrary to the prevailing T2I methods that condition their model on text embeddings $c$ by updating a hidden representation $h$, we follow a different paradigm in HyperC-GAN, and explore hypernetworks (Ha et al., 2016) that condition the model on textual information $c$ by modulating the model weights. Such a procedure can be seen as creating a different instance of the model for each conditioning vector $c$ and was recently shown (Galanti & Wolf, 2020) to be a significantly more expressive mechanism than the embedding-based conditioning approaches prevalent in existing T2I literature. A traditional hypernetwork (Chang et al., 2020) generates the entire parameter vector $\theta$ from the conditioning signal $c$ ie. $\theta = F(c)$, but this quickly becomes infeasible as the number of parameters scale. In modern neural networks, $|\theta|$ easily spans millions of parameters. Thus we develop a general approach of altering a convolutional weight tensor $W$ with a *modulating* hypernetwork $F(c)$. Our hypernetwork produces a modulation $F(c) = M$ of the same size as the weight tensor $W$ in a tensor-decomposed form. This tensor is then used to alter $W$ via an elementwise multiplication operation $W_c = W \odot M(c)$. The proposed approach is universal: we use *exactly* the same mechanism for both the generator and the discriminator, for continuous and discrete image generators and for conditioning on sentence-level or word-level signals. Typically, T2I models like AttnGAN (Xu et al., 2018), ControlGAN (Li et al., 2019a), XMC-GAN (Zhang et al., 2021), use vastly different architecture-specific ways to provide text information depending on whether it is the generator or the discriminator that is being conditioned, often along with additional text-matching losses. By comparison, the hyperconditioning mechanism in our approach is architecture agnostic. Our idea can be seen as a generalization of the mechanism used by StyleGAN2 to input noise information: in their work, the authors multiply the generator convolution weights $W$ along the input channel dimension with scales $\gamma$ corresponding to the input feature maps. In our model, 1) we use rich textual conditioning to modulate both G and D and 2) we develop a universal conditioning mechanism and demonstrate that sentence and attention based word level modulation, as well as continuous and discrete pixel-based generators can be effectively conditioned using our approach.3) HyperCGAN leverages word-level information with a novel language-guided tensor self-attention operator that modulates convolutional filters at the word level.

**Contributions.** (1) We propose a general framework, named, HyperCGAN, for synthesizing photo-realistic discrete and continuous images from text descriptions. The model is augmented with a novel language-guided mechanism that modulates convolutional filters at the word level. (2) We perform extensive qualitative and quantitative experiments validating our design choices and show the influence of the different components of our hypernetwork based language-guided modulation. (3) We show that our method has the ability to meaningfully extrapolate outside the image boundaries, and can outperform existing methods on COCO and ArtEmis datasets, including stacked generators and single generator methods. (4) We establish a strong baseline on a new affective T2I benchmark based on the recently proposed ArtEmis dataset (Achlioptas et al., 2021), which has 455,000

affective utterances collected on more than 80K artworks. ArtEmis contains captions that explain emotions elicited by a visual stimulus, learning leverages learning that can lead to more human cognition-aware T2I synthesis generation models.

## 2   RELATED WORK

**Text-to-Image Generation.** T2I synthesis has been an active area of research since at least Mansimov et al. (2015) and Reed et al. (2016a) proposed a DRAW-based (Gregor et al., 2015) model to generate images from captions. Reed et al. (2016a) first demonstrated improved fidelity of the generated images from text using Generative Adversarial Networks (GAN) (Goodfellow et al., 2014). Several GAN-based approaches for T2I synthesis have emerged since. StackGAN (Zhang et al., 2017) proposed decomposing the T2I generation into two stages - a coarse to fine approach generating a $64 \times 64$ image first and then the $256 \times 256$ image. They also proposed using conditional augmentation of the conditioning text. Later, AttnGAN (Xu et al., 2018) extended StackGAN to three stages and adopted cross-modal attention mechanisms for improved visual-semantic alignment and grounding. ControlGAN (Li et al., 2019a) improved on AttnGAN by using a channel-wise attention-driven generator to disentangle different visual attributes, allowing the model to better focus on manipulating subregions corresponding to the most relevant words. Zhu et al. (2019) proposed DM-GAN, using a memory module on the AttnGAN stacked architecture. The memory module dynamically modulates fuzzy image contents when the initial images are not well generated. AttnGAN, ControlGAN, and DM-GAN are stacked architectures augmented with the Deep Attentional Multimodal Similarity Model (DAMSM). The DAMSM loss aligns the text-image semantic based on word-region pairs. Tao et al. (2020) introduced DF-GAN that incorporates a Deep text-image Fusion block and Matching-Aware Gradient Penalty loss to encourage the visual semantic alignment. In this work we show that our hypernetwork based conditioning can beat the AttnGAN, ControlGAN, DM-GAN and DF-GAN baselines on image quality and image-text alignment. DALL-E (Ramesh et al., 2021a) is 12-billion parameter version of GPT-3 (Brown et al., 2020) trained for T2I synthesis on large scale data. Their focus is on zero-shot generation with scale and cannot be directly compared with our work. XMC-GAN (Zhang et al., 2021) recently used multiple contrastive losses to maximize the mutual information between image and text. They report significantly better FID scores, but we were unable to reproduce these scores using their repository. As a result, we do not include it as a baseline. There are non-GAN based works in T2I, e.g. (Reed et al., 2016b; 2017) which try to synthesize images condition on text information in autoregressive fashion, and (Mahajan et al., 2020) which uses an invertible-neural network for solving tasks like image captioning and T2I synthesis.

**Art generation.** Synthetically generating realistic artwork with conditional GAN is challenging due to the multiplicity of objects and unstructured-shapes in art as well as its abstract and metaphoric nature. It is less understood how capable neural representations are to learn these concepts about artistic evolution. Elgammal et al. (2018), showed that high-level internal representations of a CNN (LeCun et al., 1995) trained for art style classification can learn the Wölfflin principles of artwork. Art style is often recognized (Lecoutre et al., 2017; Tan et al., 2016) as the most challenging class to grasp and several works have explored learning artistic style representations. ArtGAN (Tan et al., 2017; 2018)] trained a conditional GAN on artist, genre, and style labels. Odena et al. (2017)] proposed an emotion to art (Alvarez-Melis & Amores, 2017) generative model by training an AC-GAN on ten classes of emotions. Another line of work includes CAN (Elgammal et al., 2017) and later H-CAN (Sbai et al., 2018), which are modified versions of GAN capable of generating creative art by learning about styles and deviating from style norms. Compared to these GANs, the hypernetwork-based conditioning we introduce in this paper can be applied to any GAN. We extend prior work in art generation by applying our conditioning to the novel text-to-continuous-image generation task. In this work, we also benchmark our model on the challenging ArtEmis (Achlioptas et al., 2021) dataset that contains captions explaining the emotions elicited by the corresponding art images. By using HyperCGAN to generate artistic images in our while being conditioned on the corresponding verbal explanations, we hope to better learn the relationship between symbolic and abstract components in human cognition aware T2I synthesis.

**Connection to HyperNetworks.** Hypernetworks are models that generate parameters for other models. Generative Hypernetworks, also called implicit generators(Skorokhodov et al., 2021a; Anokhin et al., 2021) were recently shown to rival StyleGAN2 (Karras et al., 2020). Hypernet-

works have also been applied to several tasks in architecture search (Zhang et al., 2019), few-shot learning (Bertinetto et al., 2016), and continual learning (von Oswald et al., 2020). Our HyperCGAN can generate continuous images conditioned on text using two types of Hypernetworks: (1) Image generator Hypernetworks, which produces an image represented by its implicit neural representation(INR), a neural network $F_\theta(x, y)$ that predicts an RGB pixel value given its (x,y) coordinate. (2) Text controlling Hypernetworks guides the learning mechanism of the image generator Hypernetwork with the input text. Despite the progress on unconditional INR-based decoders (e.g., Lin et al. (2019); Skorokhodov et al. (2021a); Anokhin et al. (2021); Skorokhodov et al. (2021b)), generation high-quality continuous images conditioned on text is less studied compared to discrete image generators. Our Hypernetworks augmented modulation approach facilitates conditioning the continuous image generator on text while preserving the desired INR properties (e.g., out-of-the-box-super resolution, extrapolation outside image boundaries).

## 3 APPROACH

The T2I generation task can be formulated as modeling the data distribution of images $\mathbb{P}_r$ given a conditional signal $c$. We use a standard GAN training setup where we model the image distribution using a generator $G$. In our case, $c$ is text information, sentence, or word embeddings. During training, we alternate between optimizing the generator and discriminator objectives:

$$
\begin{aligned}
L_{\mathrm{D}}(c) &= -\mathbb{E}_{x\sim\mathbb{P}_r}[-1 + D(x,c)] - \mathbb{E}_{G(z,c)\sim\mathbb{P}_g}[-1 - D(G(z,c),c)] \\
L_{\mathrm{G}}(c) &= -\mathbb{E}_{G(z,c)\sim\mathbb{P}_g} D(G(z,c),c) + \lambda L_{\mathrm{DAMSM}}
\end{aligned}
\tag{1}
$$

where $\mathbb{P}_g$ is the generated image distribution , and $\mathbb{P}_r$ is its real distribution. We also explore the role of the Deep Attentional Multimodal Similarity Model (DAMSM) based image-text matching loss $L_{\mathrm{DAMSM}}$, first proposed in Xu et al. (2018); also detailed in Appendix 6.5. Using an attention mechanism, the DAMSM is able to compute the similarity between the generated image and the sentence using both the global sentence level information and the fine-grained word level information. The DAMSM loss provides an additional fine-grained image-text matching loss for training the generator.

To facilitate both Discrete and Continuous image generation, HyperCGAN augments StyleGAN2 and INR-GAN, introduced as unconditional models, with an effective modulation mechanism that encourages better sentence-level and word-level alignment. It improves upon the image-text alignment guided by DAMSM loss while preserving high image-fidelity. In the following section, we first describe the StyleGAN2 and INR-GAN generator backbones before introducing our HyperCGAN modulation, which facilitates conditional generation based on text.

### 3.1 DISCRETE AND CONTINUOUS GENERATOR BACKBONES

**Discrete Generator: StyleGAN2** (Karras et al., 2020). StyleGAN2 is a generative model for images that achieves state-of-the-art in several distributional quality metrics. The distinguishing feature of StyleGAN2 is its unconventional generator architecture, which comprises a mapping network and a synthesis network. Instead of feeding the input latent code $z \in Z$ to only the beginning of the network, the mapping network first transforms it to an intermediate latent code $L$. Affine transform operation on $L$ then produces styles that control the layers of the synthesis network (synthesis layers) via explicit normalization and modulation. The modulation introduced in StyleGAN2 essentially scales each input convolutional feature map based on the incoming style, which can alternatively be implemented by scaling the convolution weights as,

$$
w'_{ijkl} = s_i . w_{ijkl}
\tag{2}
$$

Here, $w$ and $w'$ are the original and modulated weights, $s_i$ is the scale corresponding to $i-$th input feature map $c_{\mathrm{in}}$, $j$ enumerates the output feature maps $c_{\mathrm{out}}$, and $k$ and $l$, the spatial footprint of the convolution respectively.

**Continuous Generator: INR-GAN** (Skorokhodov et al., 2021a) Implicit Neural Representation (Skorokhodov et al., 2021a; Anokhin et al., 2020) is an unconditional approach to represent a 2D image where MLPs are used to produce RGB pixel values given image coordinate locations $(x, y)$. Similar to StyleGAN2 architecture, INR-based generator uses a mapping network and a synthesis

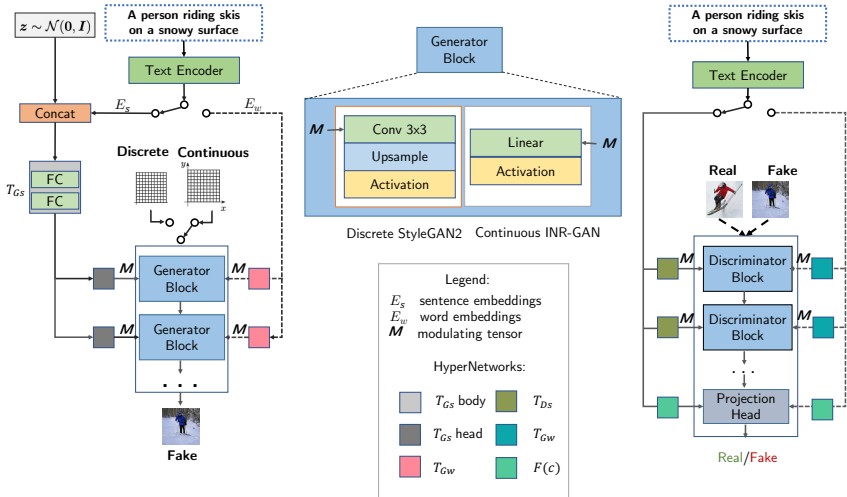

Figure 2: **The architecture of the proposed HyperCGAN** comprising of a Text encoder; hyper-networks $T_{Gs}$ body, $T_{Gs}$ and $T_{Gw}$ and heads for conditioning the generator; and $T_{Ds}$ and $T_{Dw}$ for conditioning the discriminator. HyperCGAN is a one-stage pipeline for image synthesis using continuous or discrete generators. Sentence-level conditioning hypernetworks use the $s$ subscript while the attention-based word-level conditioning heads use the $w$ subscript. The discrete Style-GAN2 generator takes a constant input, while the continuous INR-GAN generator takes an $(x, y)$ coordinate.

network. Unlike StyleGAN2, its mapping network acts as a Hypernetwork that takes a noise vector $z \sim \mathcal{N}(0, I)$ and generates parameters for an MLP model $F_{\theta(z)}(x, y)$. The MLP model becomes the synthesis network evaluated at each location of a predefined coordinate grid to synthesize an image. The weights of the MLP model are modulated through a Factorized Multiplicative Modulation (FMM) mechanism. In this mechanism, two matrices A and B, produced by the mapping network, are multiplied together, and their product is passed through an activation function to obtain a modulating tensor. Later, this modulating tensor is multiplied with the shared parameters matrix of the synthesis network. We used a multi-scale version of INR-GAN (Skorokhodov et al., 2021a), which splits $F_{\theta(z)}(x, y)$ into $K$ blocks, where earlier blocks compute low-res features that are then replicated and passed to the next level.

## 3.2 HYPER-CONDITIONAL GANS (HYPERCGANS)

Figure 2 shows an overview of over proposed HyperCGAN approach. We apply the HyperCGAN conditional modulation framework to the discrete StyleGAN2 generator and the continuous INR-GAN generator. The StyleGAN2 blocks are Convnet based and the weights are generally four dimensional ie. $M^\ell \in \mathbb{R}^{c_{out} \times c_{in} \times k_h \times k_w}$, while the INR-GAN blocks are two dimensional MLPs. ie. $M^\ell \in \mathbb{R}^{c_{out} \times c_{in} \times 1 \times 1}$. Both GAN models use the StyleGAN2 discriminator with four dimensional Convnet blocks. We condition both generator and discriminator on text embeddings $c$ which are transformed by the hypernetwork to produce modulating tensors for weight matrices. Existing GAN models for T2I generation usually exploit two kind of encodings as an input to their model: sentence encodings (Tao et al., 2020), word level encodings, or a combination of both of them (Zhu et al., 2019; Xu et al., 2018; Li et al., 2019a). Depending on the type of embeddings, we choose a different type of architectures for our hypernets as detailed below.

**Generator conditioning**: When conditioning the generator, the HyperNet backbone receives as input the concatenation of noise vector $z \sim \mathcal{N}(0, I)$ of size $512$ and sentence embedding vectors $c$ of size $d_c$. The output of the backbone is consumed by $T_{Gs}$ head layers, which then condition the generator via tensor modulation. When using sentence-level conditioning, we use an two-layer MLP for the shared backbone that we call $T_{Gs}$ body and a single-layer MLP as the output layer ($T_{Gs}$ head), which is different for each generator block. For word-level conditioning, we use Conv 1x1 layers for each block in the generator ($T_{Gw}$ block); see the Figure 2 generator part.

**Discriminator conditioning**: When conditioning the discriminator, the HyperNet backbone only receives as input either the sentence or word embedding vectors of the input sentence of size $d_c$. Note that sentence level information is also captured by our word-level modulation using a self-attention mechanism as we show later. We call the word level modulating hyper-blocks as $T_D w(c)$ and the sentence level modulating hyper-blocks as $T_D s(c)$. In case of sentence embeddings, $T_D s(c)$ blocks are single-layer fully connected neural network, but $T_D w(c)$ blocks are conv1x1 for the word-level input. Output of these blocks are used to condition the body of the discriminator: it receives the output from the backbone, and conditions the body by modulating the convolutional layers in the discriminator. To condition the final projection head in the discriminator on $c$, we use $s = h^\top F(c)$, where $h$ is the output of the last discriminator hyper-block, $s$ is the output of the discriminator, and $F(c)$, the tensor produced by our hyper-block. This form resembles the traditional Projection Discriminator (Miyato & Koyama, 2018) that uses output $s = h^\top k$, (k one-hot), which we generalize to condition on beyond one-hot class labels; see the Figure 2 discriminator part..

### 3.2.1 EXTREME MODULATING TENSOR FACTORIZATION.

Producing a full-rank modulating tensor $M^\ell$ for each block $l$ is memory-intensive and infeasible even for modestly sized architectures. For example, if the hidden layer size of our hypernetwork is of size $d_h = 512$ and the convolutional weight tensor at layer $\ell$ is of dimensionality $d_o = c_{\text{out}} \times c_{\text{in}} \times k_h \times k_w = 512 \times 512 \times 3 \times 3 \approx 2.4$ million, then the output weight matrix in the hypernetwork will be of size $d_o \times d_h = 1.2$ billion. To overcome this issue, we propose factorizing the modulating tensor with an *extreme* low-rank tensor decomposition. The canonical polyadic (CP) decomposition (Kiers, 2000) lets us express a rank-$R$ tensor $\mathcal{T} \in \mathbb{R}^{d_1 \times \ldots \times d_n}$ as a sum of $R$ rank-1 tensors:

$$\mathcal{T} = \sum_{r=1}^{R} \mathbf{t}_1^r \otimes \ldots \otimes \mathbf{t}_n^r \tag{3}$$

where $\otimes$ is the tensor product and $\mathbf{t}_r^k$ is a vector of length $d_k$. As an example in our model, our modulating tensor $M^\ell \in \mathbb{R}^{c_{\text{out}} \times c_{\text{in}} \times k_h \times k_w} = \sum_{r=1}^{R} M_r^\ell$, where $M^\ell$ is of rank $R$, and $M_r^\ell \in \mathbb{R}^{c_{\text{out}} \times c_{\text{in}} \times k_h \times k_w} = \mathbf{t}_{\text{out}}^r \otimes \mathbf{t}_{\text{in}}^r \otimes \mathbf{t}_h^r \otimes \mathbf{t}_w^r$, where each $M_r^\ell$ is composed from the tensor product of 4 vectors along the $c_{\text{out}}$, $c_{\text{in}}$, $k_h$ and $k_w$ dimensions.

**Sentence level conditioning.** Conditioning on sentence embeddings using the HyperCGAN is relatively straight-forward. The input text sentence is encoded into a finite dimensional vector using the text encoder, and then transformed into the modulating tensor $M_\ell^r$ per block $l$ via the HyperNet backbone and output layer. The modulation mechanism is simply an elementwise dot product.

### 3.2.2 WORD-LEVEL MODULATION

Since the word embeddings consists of sequence of individual word encodings, it may contain fine-grained and more grounded information. First, word encodings are passed through a hypernetwork that consists of a single conv1x1 layer (i.e., $T_{Gw}$ for the generator or $T_{Dw}$ for the discriminator). This step generates a tensor $\mathcal{W}^\ell \in \mathbb{R}^{\Omega \times (c_{\text{in}} + k_h + k_w)}$ where $\Omega$ denotes sequence length of the word embeddings (i.e., the number of words). From the word tensor $\mathcal{W}^\ell$, we derive a two dimensional matrix $Q^\ell \in \mathbb{R}^{\Omega \times (c_{\text{in}} \times k_h \times k_w)}$ using tensor factorization via an outer product operation.

$$Q_i^\ell = \mathbf{v}_{\text{in}}^i \otimes \mathbf{v}_h^i \otimes \mathbf{v}_w^i \tag{4}$$

where $Q_i^\ell$ is the $i$-th row in $Q^\ell$, $\mathbf{v}^i \in \mathbb{R}^{c_{\text{in}} + k_h + k_w}$ is the modulating hypernetwork output for the $i$-th word, $\mathbf{v}_{\text{in}}^i$ is its first $c_{\text{in}}$ dimensions, $\mathbf{v}_h^i$ is its second $k_h$ dimensions, $\mathbf{v}_w^i$ are its third $k_w$ dimensions. We apply scaled dot product attention mechanism (Vaswani et al., 2017) to attend to the relevant words in the resulting tensor $M^\ell \in \mathbb{R}^{\Omega \times c_{\text{in}} \times k_h \times k_w}$:

$$M^\ell = \text{softmax}(\frac{W^\ell (Q^\ell)^T}{\sqrt{c_{\text{out}}}}) Q^\ell, \tag{5}$$

where $W^\ell$ is the weight matrix at layer $l$, $M^\ell$ is the final modulating tensor, $W^\ell$ and $M^\ell \in \mathbb{R}^{c_{\text{out}} \times c_{\text{in}} \times k_h \times k_w}$. The proposed hypernet-based conditioning can easily be integrated to both Style-GAN2 and INR-GAN architectures. We call the resultant model architectures Hyper-conditional GANs (HyperCGAN). In INR-GAN generators, convolutional layers are replaced with linear ones whose weights are of dimension $c_{out} \times c_{in}$. In this case, $k_h, k_w$ dimensions are set to 1.

**What does Word-Level modulation mean for Discrete and Continuous Generation?** In discrete generators, our modulation operation encourages unpaired grounding between the modulating tensor of the words $Q_i^\ell, i \in 1 \cdots \Omega$ and the $c_{out}$ convolutions filters in the weight matrix of the target layer; each filter is of size $c_{in} \times k_h \times k_w$. However, in the case of continuous generations with INR-based decoders, the modulation aims at grounding words to independent pixels at an input $(x, y)$ coordinate, represented as low-res features at earlier layers and the final RGB value in the last layer.

## 4 EXPERIMENTS AND RESULTS

We perform a comprehensive evaluation of HyperCGAN on two challenging datasets and compare our HyperCGAN to state-of-the-art models including (AttnGAN, ControlGAN, DM-GAN and DF-GAN) on MS-COCO and ArtEmis datasets. MS-COCO (Lin et al., 2015) is a commonly used benchmark for T2I generation, and contains over 80K images for training and more than 40K images for testing. Each image has 5 associated captions that describe the visual content of the image. We use the splits proposed in Xu et al. (2018) to train and test our models. ArtEmis (Achlioptas et al., 2021) is a recently introduced visual art dataset built on top of WikiArt. It contains over 450K emotion attributes and explanations from humans on more than 81K artworks. Each image is associated with at least 5 captions, but unlike COCO, they are more affective and subjective than descriptive. Since most WikiArt paintings do not depict real objects, the corresponding utterances also do not reflect the visual content directly but rather through emotional statements, metaphors, and similes. These aspects of the dataset impose additional challenges on our T2I generation task. We split the dataset into 80%, 5%, 10% images for train, val, and test, respectively.

**Image Quality.** We adopt widely used automated metrics to measure the image quality of our generations: the Inception score (IS) (Salimans et al., 2016) and Frechet Inception Distance (FID) (Heusel et al., 2017). Due to the limitations of IS (Tao et al., 2020; Li et al., 2019b) to capture diversity and quality of the generation, we rely on FID as a more robust measure for T2I model evaluation like other recent related works (Zhang & Schomaker, 2020; Tao et al., 2020). We quantify the performance of our model on 30,000 images conditioned on text descriptions from the test set.

**Text-Image Alignment.** Image quality scores alone cannot reflect whether the generated image is well conditioned on the given text description. Instead, we use *R-precision* (Xu et al., 2018) for this purpose. Given a generated image, a positive caption and negatives, R-precision measures the retrieval rate for the positive caption using a surrogate multi-modal network. We follow the methodology in Zhu et al. (2019); Xu et al. (2018); Li et al. (2019a); Zhu et al. (2019), which relies on a pretrained DAMSM network for R-precision computation. However, as reported in recent work (Zhang et al., 2021; Cho et al., 2020; Zhang & Schomaker, 2020), the R-precision on real COCO images is only 22.22% . As Zhu et al. (2019); Xu et al. (2018); Li et al. (2019a); Zhu et al. (2019) models are trained to optimize for the DAMSM loss metric, they report DAMSM-based R-precision scores significantly higher than real images (see Table 1). Motivated by these findings, we decided to use the recently proposed benchmark (Park et al., 2021) based on CLIP model (Radford et al., 2021) for calculating the text-image alignment, that we term CLIP-R. Similar to Park et al. (2021), we fine-tune CLIP on both datasets. During evaluation, candidate text descriptions for each query image consists of one ground truth and 99 randomly selected mismatching descriptions. Table 1 shows that the CLIP-R on real data is significantly higher in both datasets than its counterpart DAMSM-R, suggesting that CLIP-R is more reliable metric for computing Text-Image Alignment than its DAMSM-based counterpart.

**Human evaluation.** While automated metrics are indicative, the gold standard for evaluating synthesized images is human evaluation. We conducted two evaluations on both datasets. For all human studies, 250 generations were selected at random. For each image, we assigned 5 participants in Amazon Mechanical Turk and collected 1250 responses in total.

*Visual Semantic Matching:* We compared our HyperCGAN against the DM-GAN and DF-GAN baselines. In this study, given a text description and generations in random order, subjects were asked which of the images are more consistent with the text description. If there are multiple choices, we requested them to pick the one with the most perceptually appealing visual content. Based on the collected votes, for COCO, the percentage of votes we obtained for DF-GAN, DM-GAN and HyperCGAN models were 22.88%, 32.53% and 44.59%, respectively, while on ArtEmis they were

| Model | COCO $256^2$ | | | | ArtEmis $256^2$ | | | |
|---|---|---|---|---|---|---|---|---|
| | IS ↑ | FID ↓ | DAMSM-R ↑ | CLIP-R ↑ | IS ↑ | FID ↓ | DAMSM-R ↑ | CLIP-R ↑ |
| AttnGAN | 25.89 | 34.2* | 81.52% | 29.31% | 9.01 | 45.64 | 78.68% | 7.11% |
| ControlGAN | 24.06 | 34.52* | 82.43% | 24.96% | 8.63 | 42.01 | 78.75% | 7.38% |
| DM-GAN | **30.49** | 32.64 | **88.56%** | 40.31% | **9.51** | 31.4 | **93.54%** | 12.92% |
| DF-GAN | 17.45* | 28.92 | 55.85%* | 26.13% | 7.33 | 25.4 | 52.38% | 9.81% |
| HyperCGAN$_{StyleGAN2}$ (ours) | 21.05 | **20.81** | 67.92% | **61.49%** | 7.46 | **21.7** | 27.20% | **19.86%** |
| HyperCGAN$_{INR}$ (ours) | 18.29 | 30.44 | 56.98% | 52.64% | 7.37 | 22.75 | 21.70% | 16.00% |
| Real Images | 34.88$^{\dagger}$ | - | 22.22%$^{\ddagger}$ | 89.43% | 9.93 | - | 23.87% | 45.12% |

Table 1: **Automated metrics for various models on COCO and ArtEmis** $256^2$**.** "*" denotes scores that were not reported in the original paper, but computed using either official checkpoints (when available) or by training the model using the official code. † and ‡ denote Cho et al. (2020) and Zhang & Schomaker (2020), respectively.

| Model | Conditioning | DAMSM Loss | DAMSM-R ↑ | CLIP-R ↑ | FID ↓ |
|---|---|---|---|---|---|
| StyleGAN2 | sentence | ✗ | 47.16% | 44.13% | **17.18** |
| HyperCGAN$_{StyleGAN2}$ | sentence | ✗ | 50.11% | 49.85% | 23.59 |
| HyperCGAN$_{StyleGAN2}$ | sentence | ✓ | 64.04% | 54.45% | 31.47 |
| HyperCGAN$_{StyleGAN2}$ | word | ✗ | 48.49% | 47.41% | 20.64 |
| HyperCGAN$_{StyleGAN2}$ | word | ✓ | **67.92%** | **61.49%** | 20.81 |
| INR-GAN | sentence | ✗ | 38.61% | 34.91% | 27.73 |
| HyperCGAN$_{INR}$ | sentence | ✗ | 40.36% | 40.81% | 28.29 |
| HyperCGAN$_{INR}$ | sentence | ✓ | 39.92% | 52.30% | 38.66 |
| HyperCGAN$_{INR}$ | word | ✗ | 35.67% | 37.23% | **25.39** |
| HyperCGAN$_{INR}$ | word | ✓ | **56.98%** | **52.64%** | 30.44 |

Table 2: **Ablating different HyperCGAN modulation mechanisms on COCO** $256^2$ **for Style-GAN2 and INR-GAN.** Our hypernetwork-based conditioning makes it possible to use word-level conditioning, which is crucial in achieving good results. Note that adding DAMSM loss decreases FID (i.e., overall image quality) dramatically for sentence-conditioned models.

25.6%, 29.6%, 44.8%. The results show that HyperCGAN received significantly more votes than the baselines. Results are summarized in Table 3.

*Extrapolation meaningfulness.* Since our INR-based HyperCGAN has the property of being able to extrapolate outside of the image dimensions, we were interested whether the extended regions made sense beyond the training data coordinates. In this study, we asked subjects to indicate a) whether the extended area in the generations are meaningful, and b) whether the extended area makes the image more aligned with the text description. In Table 4, for (a), the results shows that 68.8% of responses indicate that the out-of-the-region extrapolation is meaningful while 20.8% of them say it is not. For (b), 66.4% of them suggested that the alignment between the image and text description improved or remained the same (see Table 5).

**COCO Results.** Table 1 highlights results of baselines and our proposed model on automated metrics. On COCO dataset, HyperCGAN$_{StyleGAN2}$ achieves the best FID score while HyperCGAN$_{INR}$ being comparable to the baselines, 20.81 and 30.44, respectively. In terms of retrieval scores, all the baselines except DF-GAN gives DAMSM-R scores higher than 80%. However, they underperform in terms of the CLIP-R metric. This can be explained by the fact that these models are optimized for DAMSM-R metric using DAMSM-loss during training. Our models, however, achieve higher CLIP-R precision or perform at par with DM-GAN, meaning that generations from HyperCGANs are more aligned with textual information. This is supported by our human studies, where 44.59%

Table 3: Visual Semantic Matching Results.

Table 4: Extrapolation meaningfulness.

Table 5: Extrapolation alignment with text.

| Dataset | DM-GAN | DF-GAN | HyperCGAN |
|---|---|---|---|
| COCO | 32.53% | 22.88% | 44.59% |
| ArEmis | 29.60% | 25.60% | 44.80% |

| Dataset | meaningful | not sure | not meaningful |
|---|---|---|---|
| COCO | 68.8% | 10.4% | 20.8% |
| ArEmis | 75.6% | 8% | 16.4% |

| Dataset | more aligned | same | less aligned |
|---|---|---|---|
| COCO | 14.4% | 52% | 33.6% |
| ArEmis | 12.8% | 56.4% | 30.8% |

of responses from participants indicate that HyperCGAN generations are more consistent with the text description.

**Affective Text-to-Image Synthesis.** Since ArtEmis contains affective text descriptions, it introduces additional challenges to the T2I generation task. We see in Table 1, that HyperCGAN achieves 21.70 FID score and outperforms all baselines. The trend in R-prec scores is similar to the COCO dataset, where HyperCGAN underperforms the other baselines on R-prec, but overperforms them on CLIP-R. On visual inspection of the generated images in Figure 3, we see that the HyperCGAN generated artwork abounds in vivid, high-quality details. It precisely captures major scenes when it comes to depicting water surfaces, open space landscapes, cities and architecture, human figures and their postures, also shows the diversity of paintings.

**Ablations on hyper-modulation mechanisms.** Table 2 shows that simply conditioning generators with our hyper-modulation method achieves comparable results both in terms of image-quality and text-image alignment metrics. Although our models perform well without using DAMSM loss, we also investigate HyperCGAN performance with DAMSM loss. Results indicate that DAMSM-Rprec not only increases while training with this type of loss but CLIP-R score also improves significantly with our proposed word-level modulation mechanism. On COCO dataset, we can observe that HyperCGAN$_{StyleGAN2}$ using DAMSM loss improved with word-level modulation from 54.41% to 61.49% on CLIP-R and from 31.47 to 20.81 on FID score. Similarly, HyperCGAN$_{INR}$ using DAMSM loss improved with word-level modulation from 52.3% to 52.64% on CLIP-R and from 38.66 to 30.44 on FID score.

**Ablations on Modulating Tensors.** We further investigate the effect of rank and modulation dimension of the modulating tensors in the architecture of HyperCGAN$_{StyleGAN2}$. Specifically, we generate the modulating tensors using different combinations of rank-1 tensors $(c_{out}, c_{in}, k_h, k_w)$ through tensor factorization to modulate the weight matrices of convolution layers of both generator and discriminator. In Table 6 in the appendix, we observe that the model achieves the highest FID score when modulating the tensor along dimension $c_{in}$. However, in terms of CLIP-R score tensor of size $c_{out} \times c_{in}, k_h \times k_w$ holds the best score. We also experimented with generating modulating tensors of higher ranks. However, using higher rank modulating tensors did not improve the results, and only contributed to the complexity of the model.

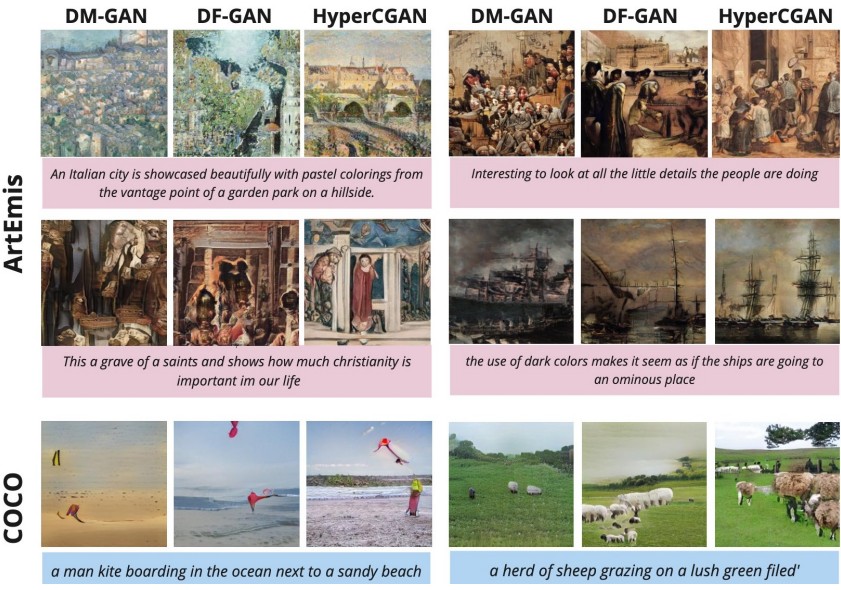

Figure 3: Qualitative comparison of the proposed HyperCGAN model with state-of-the-art models (DM-GAN, DF-GAN) for T2I synthesis.

## 5 CONCLUSION

In this paper, we propose HyperCGAN, a novel HyperNet-based conditional GAN. HyperCGAN is a model backbone-agnostic text-to-image generative model with a single generator that operates with a novel language-guided tensor modulation operator for sentence-level and word-level conditioning. HyperCGAN-based conditioning is a flexible framework and can be applied to both discrete pixel-based generators like StyleGAN2 and continuous image generators like INR-GAN as we demonstrated. Using several automated metrics for image quality, visual-semantic consistency, and human evaluation scores, we show that HyperCGAN achieves high performance in generation quality and diversity compared to existing text-to-image synthesis baselines. To our knowledge, HyperCGAN is also the first approach that facilitates text to continuous image generation, and we show its ability to meaningfully extrapolate images beyond training image dimension while maintaining the alignment with the input language description. We hope that our method may encourage future work on hyper networks for controllable conditional generation.

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

# 6 APPENDIX

| Dimension | FID | CLIP-R |
|---|---|---|
| $c_{\text{in}}$ | 18.74 | 45.13% |
| $c_{\text{out}}$ | 54.6 | 30.76% |
| $c_{\text{out}}, c_{\text{in}}$ | 21.11 | 47.86% |
| $c_{\text{in}}, k_h, k_w$ | 20.32 | 49.42% |
| $c_{\text{out}}, k_h, k_w$ | 23.49 | 43.95% |
| $c_{\text{out}}, c_{\text{in}}, k_h, k_w$ | 23.59 | 49.85% |

Table 6: Effect of different choices of modulating tensors.

## 6.1 IMPLEMENTATION DETAILS

Our models are trained with learning rate $lr = 0.0025$ with multi-gpu support on 4 NVIDIA TESLA V100 GPUs. For all experiments, we kept the batch size equal to 16 and run for 25k iterations. For COCO datasets, we followed standard splits, but we split the ArtEmis dataset into train/val/test splits in a ratio of 0.85, 0.10, 0.05. At inference, we used only test split to generate art images.

## 6.2 SENTENCE-LEVEL INFORMATION

Similar to Xu et al. (2018); Li et al. (2019a); Zhu et al. (2019); Tao et al. (2020), first we extract 256-dimensional sentence embeddings denoted as $c$ from LSTM-based pretrained text encoder. Then, we concatenate extracted embeddings and noise vector $z$ of dimension 512, and pass it through a hypernetwork $T_G(z, c)$ of the generator.

## 6.3 WORD-LEVEL INFORMATION

In order to leverage word-level information, we extract the word embeddings from the same text encoder mentioned above. However, word embeddings have different sequence lengths and not suitable for batch processing. Therefore, the words embeddings are padded with 0s matching the max word length. Then, the padded embeddings go through hypernetworks with a single conv1x1 layers to generate style vectors of dimension $\tau \times (c_{\text{in}} + k_w + k_h)$ (See Figure 9).

## 6.4 TEXT ENCODER

For text encoder, we adopt pretrained text encoder from AttnGAN. This text encoder is used in all the baselines reported in the paper. Therefore, for consistency, we also used AttnGAN text encoder

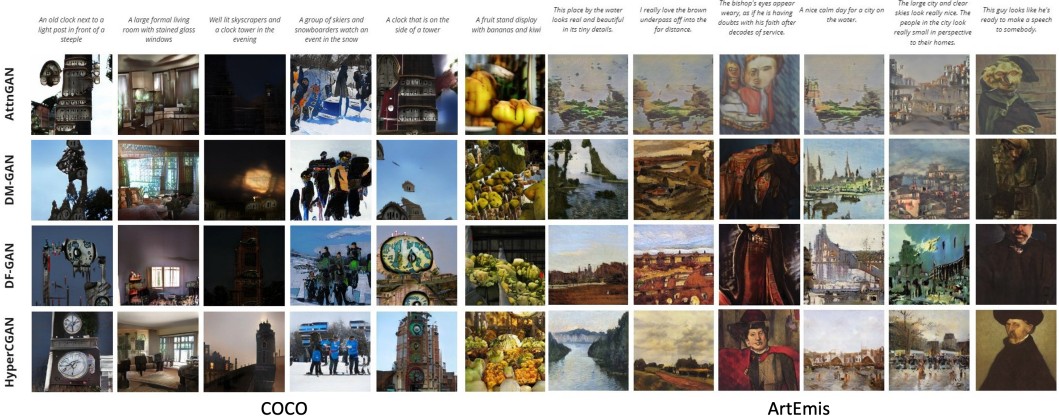

Figure 4: Qualitative Results on ArtEmis dataset (Attn-GAN, DM-GAN, DF-GAN, HyperCGAN )

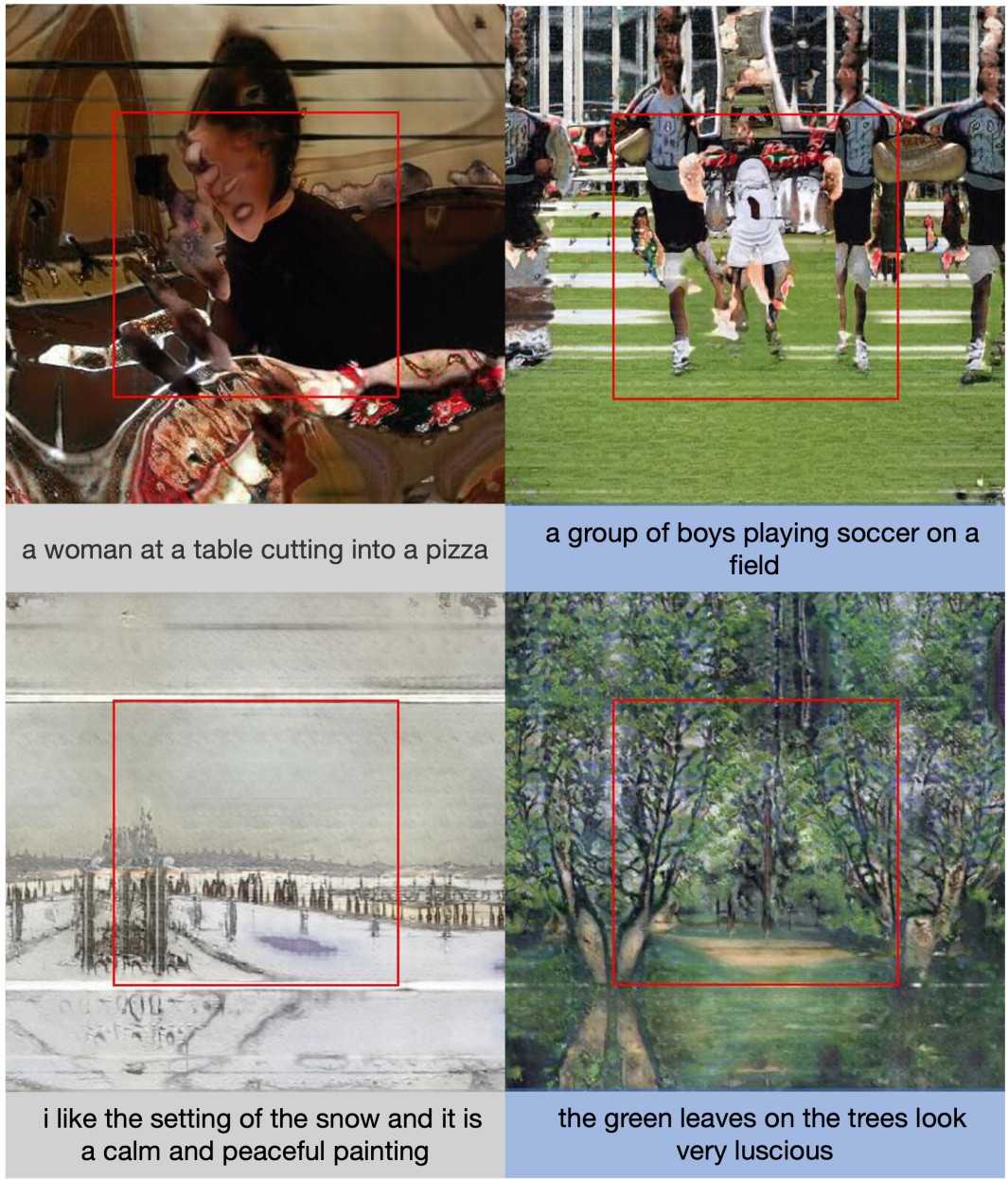

Figure 5: Out-of-the-Box generation

which is based on a bi-directional Long Short-Term Memory (LSTM). In the bi-directional LSTM, each word corresponds to two hidden states, one for each direction. To represent the semantic meaning of a word, they concatenate its two hidden states. The last hidden states of the bi-directional LSTM are concatenated to be the global sentence vector. The hidden size of both embeddings is equal to 256.

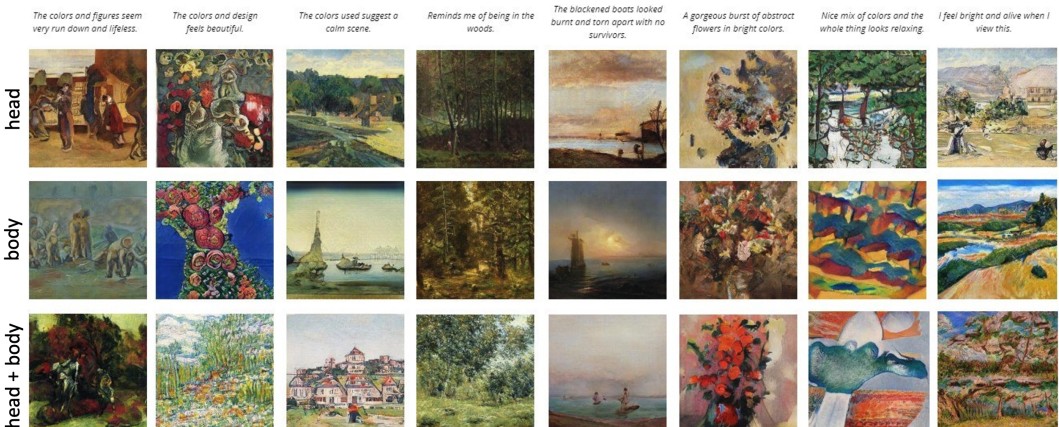

Figure 6: HyperCGAN Head and Body Ablation Qualitative Results on ArtEmis dataset

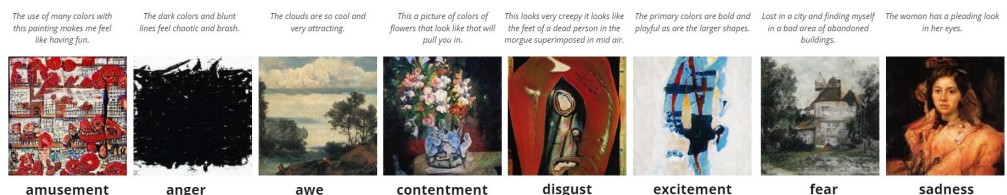

Figure 7: Example of affective captions and corresponding emotion from ArtEmis dataset.

```python
def tensor_modulation(weight, styles):
    """
    Performs a low-rank tensor modulation
    weight: [c_out, c_in, kw, kh] - weights of conv layer
    styles: [b, rank * (c_out + c_in + kw + kh)] - output of a hypernetwork
    """

    c_out, c_in, kw, kh = weight.shape
    b = styles.shape[0]
    rank = styles.shape[1] // (c_out + c_in + kh + kw)

    styles = styles.reshape(b, rank , c_out + c_in + kh + kw)
    # extract factors for tensor modulation
    factor1 = np.expand_dims(styles[:, :, : c_out].reshape(b, rank, c_out), [-3, -2, -1])
    factor2 = np.expand_dims(styles[:, :, c_out : c_out + c_in].reshape(b, rank, c_in), [2 ,-2, -1])
    factor3 = np.expand_dims(styles[:, :, c_out + c_in:c_out + c_in + kh].reshape(b, rank, kw), [2 ,-3, -1])
    factor4 = np.expand_dims(styles[:, :, c_out + c_in + kh:].reshape(b, rank, kh), [2 ,-3, -2])

    # obtain modulating tensor by low-rank tensor factorization
    modulating_tensor = factor1 * factor2 * factor3 * factor4 # [b, rank, c_out, c_in, kh, kw]
    modulating_tensor = modulating_tensor.sum(axis=1) / np.sqrt(rank) # [b, c_out, c_in, kh, kw]

    # Normalize the variance of 4 factors product with mean=std=1 each (assuming independence)
    modulating_tensor = modulating_tensor / np.sqrt(15)

    return modulating_tensor
```

Figure 8: Numpy-like pseudocode for core tensor modulation implementation.

```python
def tensor_modulation_word(weight, styles)
    """
    Performs a low-rank tensor modulation based
    on word level information
    weight: [c_out, c_in, kh, kw]
    styles: [b, c_in + kh + kw, num_words]
    """
    c_out, c_in, kw, kh = weight.shape
    b = styles.shape[0]

    styles = np.transpose(styles, (2, 1)) # [b, num_words, c_in + kh + kw]
    n_words = styles.shape[1]

    factor1 = np.expand_dims(styles[:, :, :c_in].reshape(b, n_words, c_in), [2, -2, -1])
    factor2 = np.expand_dims(styles[:, :, c_in:c_in + kw].reshape(b, n_words, kw), [2, -3, -1])
    factor3 = np.expand_dims(styles[:, :, c_in + kw:].reshape(b, n_words, kh), [2, -3, -2])

    M = factor1 * factor2 * factor3 # [b, num_words, c_in,  kh,  kw]
    M = M.reshape(b, n_words, -1) # [b, num_words, c_in x kh x kw]

    weight = np.tile(weight,(b, 1, 1, 1, 1)) # [b, c_out, c_in, kh, kw]

    weight = weight.view(b, c_out, -1) # [b, c_out, c_in x kh x kw]
    score = (weight * np.transpose(M, (2, 1))) / np.sqrt(c_out)  # [b, c_out, num_words]
    attn = softmax(score, -1)
    modulation = attn * M # [b, c_out, c_in x kh x kw]

    modulation = modulation.reshape(b, c_out, c_in, kw, kh)
    weight = weight.reshape(b, c_out, c_in, kw, kh)

    return weight * modulation
```

Figure 9: Numpy-like pseudocode for attention-based word-level tensor modulation.

## 6.5 DAMSM LOSS

DAMSM loss Xu et al. (2018) is defined on top of Inception-v3 image model Szegedy et al. (2016), which is used to extract image features $f \in \mathbb{R}^{768 \times 289}$ (reshaped from 768×17×17). 768 is the dimension of the local feature vector, and 289 is the number of sub-regions in the image. These features are then converted to a common semantic space of text features by adding an FC layer $v = Wf$, $\overline{u} = \overline{W}\overline{f}$, where $v_i$ is the visual feature vector for the $i^{th}$ sub-region of the image; and $\overline{u} \in \mathbb{R}^D$ is the global vector for the whole image. We then calculate the similarity matrix for all possible pairs of words in the sentence and sub-regions in the image by

$$s = e^T v, \tag{6}$$

where $s$ is a similarity matrix between all word-region paris, $s_{i,j}$ is the dot-product similarity between the $i^{th}$ word of the sentence and the $j^{th}$ sub-region of the image. We find that it is beneficial to normalize the similarity matrix as follows

$$\overline{s}_{i,j} = \frac{\exp(s_{i,j})}{\sum_{k=0}^{T-1} \exp(s_{k,j})}. \tag{7}$$

Then, region-context vector $c_i$ is defined as a representation of the image's sub-regions related to the $i^{th}$ word of the sentence. It is computed as the weighted sum over all regional visual vectors, i.e.,

$$c_i = \sum_{j=0}^{288} \alpha_j v_j, \quad \text{where } \alpha_j = \frac{\exp(\gamma_1 \overline{s}_{i,j})}{\sum_{k=0}^{288} \exp(\gamma_1 \overline{s}_{i,k})}. \tag{8}$$

Then, the relevance between the $i^{th}$ word and the image using the cosine similarity between $c_i$ and $e_i$, i.e., $R(c_i, e_i) = (c_i^T e_i)/(||c_i|| ||e_i||)$. The *attention-driven image-text matching score* between the entire image ($q$) and the whole text description ($d$) is defined as

$$R(q, d) = \log \Big( \sum_{i=1}^{T-1} \exp(\gamma_2 R(c_i, e_i)) \Big)^{\frac{1}{\gamma_2}}, \tag{9}$$

; we used the default parameters in Xu et al. (2018).

The DAMSM loss is finally defined as

$$\mathcal{L}_{DAMSM} = \mathcal{L}_1^w + \mathcal{L}_2^w + \mathcal{L}_1^s + \mathcal{L}_2^s. \tag{10}$$

where

$$\mathcal{L}_1^w = -\sum_{i=1}^{M} \log P(d_i|q_i), \mathcal{L}_2^w = -\sum_{i=1}^{M} \log P(q_i|d_i), \tag{11}$$

where 'w' stands for "word", where $P(q_i|d_i) = \frac{\exp(\gamma_3 R(q_i,d_i))}{\sum_{j=1}^{M} \exp(\gamma_3 R(q_j,d_i))}$ is the posterior probability that sentence $d_i$ is matched with its corresponding image $q_i$. If we redefine Eq. 9 by $R(q, d) = (\overline{v}^T \overline{e})/(||\overline{v}|| ||\overline{e}||)$ and substitute it to Eq. 12 and 11, we can obtain loss functions $\mathcal{L}_1^s$ and $\mathcal{L}_2^s$ (where 's' stands for "sentence") using the sentence vector $\overline{e}$ and the global image vector $\overline{v}$. The DAMSM loss is designed to learn the attention model in a semi-supervised manner, in which the only supervision is the matching between entire images and whole sentences (a sequence of words). Similar to Fang et al. (2015); Huang et al. (2013), for a batch of image-sentence pairs $\{(q_i, d_i)\}_{i=1}^{M}$, the posterior probability of sentence $d_i$ being matching with image $a_i$ is computed as

$$P(D_i|Q_i) = \frac{\exp(\gamma_3 R(Q_i, D_i))}{\sum_{j=1}^{M} \exp(\gamma_3 R(Q_i, D_j))}, \tag{12}$$

where $\gamma_3$ is a smoothing factor determined by experiments. In this batch of sentences, only $d_i$ matches the image $q_i$, and treat all other $M - 1$ sentences as mismatching descriptions. The loss function is defined as as the negative log posterior probability that the images are matched with their corresponding text descriptions (ground truth), as shown in Eq. 11.

## 6.6 ADDITIONAL RESULTS ON CUB DATASET

We performed additional experiments where we trained our models on CUB dataset and compared to recent baselines (DF-GAN, DM-GAN). In Table 7, the results indicate that our models achieve high CLIP-R precision and comparable FID score.

| Model | Conditioning | DAMSM Loss | DAMSM-R ↑ | CLIP-R ↑ | FID ↓ |
|---|---|---|---|---|---|
| DM-GAN | | ✓ | 75.89% | 11.98% | 16.09 |
| DF-GAN | | ✓ | 39.05% | 10.29% | 14.81 |
| HyperCGAN$_{StyleGAN2}$ | sentence | ✗ | 25.51% | 15.82% | 13.28 |
| HyperCGAN$_{StyleGAN2}$ | sentence | ✓ | 28.53% | 19.07% | 11.72 |
| HyperCGAN$_{StyleGAN2}$ | word | ✗ | 25.55% | 19.94% | 11.81 |
| HyperCGAN$_{StyleGAN2}$ | word | ✓ | 68.53% | 21.45% | 15.02 |
| HyperCGAN$_{INR}$ | sentence | ✗ | 18.43% | 17.85% | 20.82 |
| HyperCGAN$_{INR}$ | sentence | ✓ | 44.15% | 26.94% | 38.66 |
| HyperCGAN$_{INR}$ | word | ✗ | 17.46% | 17.95% | 18.73 |
| HyperCGAN$_{INR}$ | word | ✓ | 46.39% | 31.90% | 15.72 |
| Real | | | 18.24% | 25.47% | |

Table 7: **Results on CUB dataset.**

## 6.7 GENERATING HIGH RESOLUTION IMAGES (COCO)

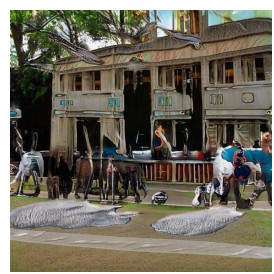

a house being built with lots of wood

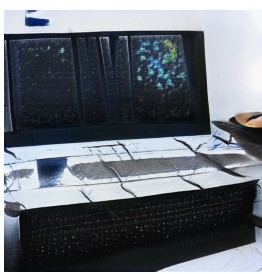

a laptop computer sits on a computer desk next to a mouse

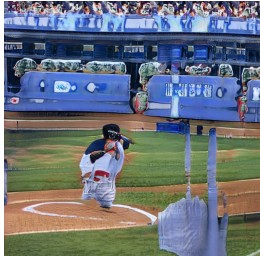

a batter backs up as the ball is thrown

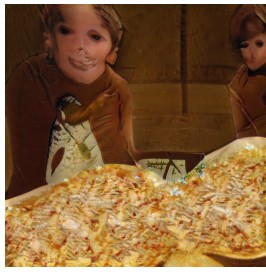

a man holding a fully topped pizza in front of the camera

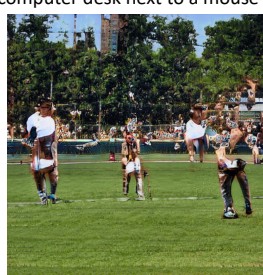

soccer players are running after the ball together

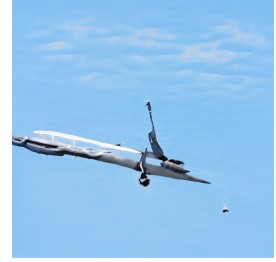

a plane is flying high in the very cloudy sky

Figure 10: **High-resolution generations (1024x1024) from our HyperCGAN$_{StyleGAN2}$ model trained on COCO.**