# OpenReview forum: "HyperCGAN: Text-to-Image Synthesis with HyperNet-Modulated Conditional Generative Adversarial Networks"
_ICLR.cc/2022/Conference — ICLR 2022 Submitted_

### Official Review · Reviewer_rpKE · 2021-10-20

**Correctness:** 3
**Technical Novelty And Significance:** 3
**Empirical Novelty And Significance:** 3
**Recommendation:** 5
**Confidence:** 4

**Main Review:**

**Strengths**

* The authors achieve state-of-the art performance on FID, their newly proposed CLIP-R metric and human evaluations using the StyleGAN-V2 backbone.
* *Auxiliary contribution*: The paper is the first application of continuous generative models for text-to-image synthesis allowing text-conditioned image extrapolation.
* As far as I can see, the paper is also the first to augment StyleGAN for text-to-image synthesis. If this is true, the authors should mention this in the introduction.

**Weaknesses**

The paper is unclear on some necessary details / simple baselines. I am open to adjusting my rating if the authors can answer them convincingly and make the corresponding changes to the manuscript.

* Is the improvement from the baselines in Table 1 due to the StyleGAN v2 backbone or the proposed hyper-network based modulation? **Experiment Suggestion** : Run the channel-wise modulation (instead of the proposed modulation) as described in Eq 2). $s_i$ being a function of both the noise variable $z$ and the text embedding $c$. Report the results in Table 1.
* The writing in Section 3.2 and Table 2 suggests that only word level or sentence level modulation is applied at a given instance. However in Figure 2, it looks as if both word and sentence level modulation are applied to the generator block. The authors should modify Figure 2, to show that only either of these is applied.
* “To condition the final projection head in the discriminator on TDs(c), we use s = h >F(c), where h is the output of the last discriminator hyper-block, s is the output of the discriminator, and F(c), the tensor produced by our hyper-block.” Why is this done only for the sentence-level modulations?. Please also add dimensionalities to the corresponding variables.
* In Section 3.2.1 “Extreme Modulating Tensor Factorization”, what are the outputs of the hypernetwork? Are they 4*R tensors ($t_1^1  \dots t_n^1 \dots t_1^R  \dots t_n^R$ )? What is R set to be? How is it ensured that the outputs have their corresponding ranks during the course of training? Please carify these details.
* Section 3.2.2 "Word-Level Modulation" can be motivated better. For, eg it seems like it is a cross-attention layer between the convolutional weights and the sequence. I suggest to run an ablation, removing this layer to showcase its importance.
* INR GAN has both a generator network and a MLP. Is the proposed modulation applied only to the MLP? Why?
* What architecture is the Text Encoder? How is the output pooled from shape (sequence length, hidden size) to (hidden size,) to form the sentence embedding? Is this trained end-to-end?
* In the human evaluations, are the images from DF-GAN, DM-GAN and HyperCGAN shown at the same time or one after the other. In either case, what is the amount of time the human rater gets between two sets of images? Please add these details to make the study reproducible. Also consider adding them to Table 1.

**Related Work**

I recommend that the authors cite other (non-GAN based) related text-to-image synthesis works.

[1] Reed et al Parallel Multiscale Autoregressive Density Estimation, ICML 2017

[2] Reed et al Generating Images with Controllable Structure ICLR Workshops 2017

[3] Mahajan et al Latent Normalizing Flows for Many-to-Many Cross-Domian Mappings ICLR 2020

[4] Liang et al CPGAN: Full-Spectrum Content-Parsing Generative Adversarial Networks for Text-to-Image Synthesis

[5] Mahajan et al Diverse Image Captioning with Context-Object Split Latent Spaces ICLR 2020

**Other details**

* It will be nice if the DAMSM loss is described in the paper to make it more self contained.
* “When using sentence-level conditioning, we use an eightlayer MLP for the shared backbone that we call TGs body and a single-layer MLP as the output layer (TGs head), which is different for each generator block. For word-level conditioning, we use separate Conv 1x1 layers for each block in the generator (TGw block); see the Fig. 2 generator part”: A single-layer MLP and a Conv 1x1 for each block in the generator is equivalent?
* Consider summarizing the “extrapolation meaningfulness results in a new table.

**Typos**

* The output of the backbone is consumed by the output layers, which then condition the generator via tensor modulation. What does “output layers” refer to?
* For all human studies, 250 generations were selected at random. For each image we assigned 5 participants in Amazon Mechanical Turk and collected 1500 responses in total. I suppose this should be 1250?


**Summary Of The Paper:**

The paper proposes a hyper-network based conditional approach for text-to-image synthesis. A hypernetwork takes as input the text representation and outputs a 4-D variable. This modulates the convolutions of both the generator and discriminator of the Generative Adversarial Network pointwise. Since predicting a 4-D Tensor is impractical, the hypernetwork predicts a low-rank decomposition, from which the 4-D variable can be constructed via tensor products.

The authors showcase this approach on a StyleGAN v2 backbone for discrete image synthesis and INR-GAN backbone for continuous image synthesis on COCO, CUB and ArtEmis Dataset.


**Summary Of The Review:**

Please see the strengths and weaknesses.

I have given this paper 5 (instead of 3) solely because of the empirical results.

I have given this paper 5 (instead of 6 or 7) because the importance of the proposed pointwise modulation is unclear emprically.
The authors should report the improvements of their proposed hyper-network based condiioning over a simple StyleGAN-style channel-wise modulation. (ideally controlling for same number of parameters / FLOPS)

Additionally, some architectural and implementation details aren't clear enough, which I have highlighted in the weaknesses section.

**Rebuttal Update**

I read the authors response. Unfortunately, I have to stick to my lean reject decision.

For a purely empirical paper, the authors have not convinced me their hyper-network based modulation is better than a simple channel-wise StyleGAN2 style modulation.

In these experiments in the rebuttal, https://openreview.net/forum?id=z-5BjnU3-OQ&noteId=-oi3O8JQB-H, the authors have shown that thier model beats a StyleGAN2 baseline with sentence conditioning on CLIP-R. But they have not run their StyleGAN2 baseline with word conditioning / DAMSM loss, which makes me a bit sceptic about the gains with the hyper-network conditioning framework as opposed to just using the StyleGAN2 baseline.

---

> ### Author Response · Authors · 2021-11-20
> **Part 1: Response to Reviewer rpKE**
>
> Thank you for your detailed review, questions, and suggestions. We here address them and incorporate all feedback.
>
> >Is the improvement from the baselines in Table 1 due to the StyleGAN v2 backbone or the proposed hyper-network based modulation? Experiment Suggestion : Run the channel-wise modulation (instead of the proposed modulation) as described in Eq 2).
>  being a function of both the noise variable  and the text embedding . Report the results in Table 1.
>
> Here we report the result of standard conditioning using our StyleGAN2 and INRGAN backbones without our method. While we observe that the image quality of these baselines is a little better in terms of FID score, they significantly underperform in R-prec metrics indicating the lack of relevance of the generated image to the input text. This shows that standard conditioning does not work well on these architectures and proper design of the conditioning learning signal is necessary to achieve good performance. The table below also shows that existing backbones benefit from our proposed word level self-attention modulation (Eq 5 in the paper), which is at the heart of our paper.
>
> | Model         | sent | word | damsm loss | R      | CLIP-R | FID     |
> |---------------|------|------|------------|--------|--------|---------|
> | **StyleGAN2**     | x    |      |            | 47.16% |  44.13 | 17.18 |
> | HyperCGAN_SG2 | x    |      |            | 50.11% |  49.85 |   23.59 |
> | HyperCGAN_SG2 | x    |      | x          | 64.04% |  54.45 |   31.47 |
> | HyperCGAN_SG2 |      | x    |            | 48.49% |  47.41 |   20.64 |
> | HyperCGAN_SG2 |      | x    | x          | 67.92% |  61.49 |   20.81 |
> | **INR Base**      | x    |      |            | 38.61% |  34.91 |   27.73 |
> | HyperCGAN_INR | x    |      |            |  40.36 |  40.81 |   28.29 |
> | HyperCGAN_INR | x    |      | x          |  39.92 |   52.3 |   38.66 |
> | HyperCGAN_INR |      | x    |            |  35.67 |  37.23 |   25.39 |
> | HyperCGAN_INR |      | x    | x          |  56.98 |  52.64 |   30.44 |
>
> Additionally, we report the number of parameters for the baselines and our models. Apart from DFGAN,  AttnGAN, ControlGAN and DMGAN baselines have 3 discriminators, adopting a multistage generation pipeline. However, our method employs only a single modulated discriminator, adopting a single-stage generation pipeline similar to DFGAN.  Here, we summarize the model parameters for each of the methods (baselines) in our paper:
>
> | Model                           | G          | D1         | D2         | D3          | Total number of parameters in D | FID     |
> |---------------------------------|------------|------------|------------|-------------|---------------------------------|---------|
> | AttnGAN                          | 13,811,456 | 13,304,450 | 42,800,258 | 160,774,274 | 216,878,982                     | 17.1795 |
> | ControlGAN                      | 36,733,888 | 13,304,450 | 42,800,258 | 160,774,274 | 216,878,982                     |   23.59 |
> | DM-GAN                          | 22,335,314 | 1,878,242  | 5,155,810  | 18,264,546  | 25,298,598                      |   31.47 |
> | DF-GAN                          | 12,240,010 | 20,170,566 |            |             | 20,170,566                      |   20.64 |
> | HyperCGAN StyleGAN2 (word)      | 29,661,154 | 26,331,649 |            |             | 26,331,649                      |   20.81 |
> | HyperCGAN  StyleGAN2 (sentence) | 25,923,504 | 25,949,156 |            |             | 25,949,156                      |   27.73 |
> | HyperCGAN_INR                   | x          |            |            |       40.36 |                           40.81 |   28.29 |
> | HyperCGAN_INR                   | x          |            | x          |       39.92 |                            52.3 |   38.66 |
> | HyperCGAN_INR                   |            | x          |            |       35.67 |                           37.23 |   25.39 |
> | HyperCGAN_INR                   |            | x          | x          |       56.98 |                           52.64 |   30.44 |
>
> [1] Karras, Tero, et al. "Analyzing and improving the image quality of stylegan." Proceedings of the IEEE/CVF Conference on Computer Vision and Pattern Recognition. 2020.
>
> >The writing in Section 3.2 and Table 2 suggests that only word-level or sentence level modulation is applied at a given instance. However, in Figure 2, it looks as if both word and sentence level modulation are applied to the generator block. The authors should modify Figure 2, to show that only either of these is applied.
>
> Thanks, we modified Figure 2 to clarify that; please see the new Figure in the attached updated version

---

> > ### Comment · Reviewer_rpKE · 2021-11-24
> > **Followup Questions**
> >
> > Thanks for the additional experiments.
> >
> > 1. HyperCGAN_S2 + Word Conditioning + DAMSM loss seems to perform the best on the Clip-R metric in the hyperCGAN variants. Can the authors please also run StyleGAN2 + Word Conditioning + DAMSM / StyleGAN2 + Sentence Conditioning + DAMSM loss as the baseline?
> > 2. Please also report training times (and/or) number of parameters of StyleGan2, since HyperCGAN adds more complexity in terms of the conditoning mechanism.
> > 3. Consider adding to the caption in Figure 2, "At a given instance, only either of sentence or word level conditioning is applied."

---

> > > ### Author Response · Authors · 2021-11-29
> > > **Re : Followup Questions**
> > >
> > > >HyperCGAN_S2 + Word Conditioning + DAMSM loss seems to perform the best on the Clip-R metric in the hyperCGAN variants. Can the authors please also run StyleGAN2 + Word Conditioning + DAMSM / StyleGAN2 + Sentence Conditioning + DAMSM loss as the baseline?
> > >
> > > We have launched experiments, but they did not fully converge yet. We will report the results as soon as possible within 10 hours
> > >
> > > >Please also report training times (and/or) number of parameters of StyleGan2, since HyperCGAN adds more complexity in terms of the conditoning mechanism.
> > >
> > > In the table below, we report number of parameters for all baselines including StyleGAN2 and INR-GAN models.
> > >
> > > | Model                          | G          | D1         | D2         | D3          | Total number of parameters in D |
> > > |--------------------------------|------------|------------|------------|-------------|---------------------------------|
> > > | AttnGAN                        | 13,811,456 | 13,304,450 | 42,800,258 | 160,774,274 | 216,878,982                     |
> > > | ControlGAN                     | 36,733,888 | 13,304,450 | 42,800,258 | 160,774,274 | 216,878,982                     |
> > > | DM-GAN                         | 22,335,314 | 1,878,242  | 5,155,810  | 18,264,546  | 25,298,598                      |
> > > | DF-GAN                         | 12,240,010 | 20,170,566 |            |             | 20,170,566                      |
> > > | HyperCGAN StyleGAN2 (word)     | 29,661,154 | 26,331,649 |            |             | 26,331,649                      |
> > > | HyperCGAN StyleGAN2 (sentence) | 25,923,504 | 25,949,156 |            |             | 25,949,156                      |
> > > | HyperCGAN_INR (sentence)       | 34,878,073 | 30,408,192 |            |             | 30,408,192                      |
> > > | HyperCGAN_INR (word)           | 36,004,473 | 29,495,937 |            |             | 29,495,937                      |
> > > | StyleGAN2                      | 23,585,250 | 24,920,128 |            |             | 24,920,128                      |
> > > | INR-GAN                        | 34,878,073 | 24,920,128 |            |             | 24,920,128                      |
> > >
> > > >Consider adding to the caption in Figure 2, "At a given instance, only either of sentence or word level conditioning is applied."
> > >
> > > Thanks for the suggestion. We will add this in the final version

---

> > > > ### Author Response · Authors · 2021-11-30
> > > > **Re : Additional Experiments**
> > > >
> > > > >HyperCGAN_S2 + Word Conditioning + DAMSM loss seems to perform the best on the Clip-R metric in the hyperCGAN variants. Can the authors please also run StyleGAN2 + Word Conditioning + DAMSM / StyleGAN2 + Sentence Conditioning + DAMSM loss as the baseline?
> > > > We launched two experiments:
> > > > 1.StyleGAN2 + Sentence Embeddings + DAMSM loss
> > > > 2.StyleGAN2 + Word Embeddings + DAMSM loss, where we took the average of word embeddings as a conditional signal.
> > > > Results are reported in the table below (new experiments are highlighted as bold in the first column).
> > > >
> > > > |     Model     | sent | word | damsm loss |    R   | CLIP-R |   FID   |
> > > > |:-------------:|:----:|:----:|:----------:|:------:|:------:|:-------:|
> > > > |   StyleGAN2   |   x  |      |            | 47.16% |  44.13 | **17.1795** |
> > > > | HyperCGAN_SG2 |   x  |      |            | 50.11% |  49.85 |  23.59  |
> > > > |  **StyleGAN2**   |   x  |      |      x     | 68.05% |  52.62 |  33.61  |
> > > > | HyperCGAN_SG2 |   x  |      |      x     | 64.04% |  54.45 |  31.47  |
> > > > | HyperCGAN_SG2 |      |   x  |            | 48.49% |  47.41 |  20.64  |
> > > > |  **StyleGAN2**   |      |   x  |      x     | 66.24% |  57.77 |  19.21  |
> > > > | HyperCGAN_SG2 |      |   x  |      x     | 67.92% | **61.49** |  20.81  |
> > > >
> > > >  Our HyperCGAN_SG2 + sentence + damsm loss archives lower scores in FID (31.47) and higher CLIP-R (54.45) compared to StyleGAN2 + sentence + damsm loss. The FID score degrades in both setups when we use DAMSM loss.
> > > > For StyleGAN2 + Word Embeddings + DAMSM loss, even though it achieves slightly better FID score (19.21) compared to our HyperCGAN_SG2 + word + DAMSM loss (20.81), however, it underperforms in both retrieval scores achieving 66.24% and 57.77% in DAMSM-R and CLIP-R respectively. Our model outperforms StyleGAN2 + word + DAMSM loss by 3.72% in CLIP-R, which is more reliable in terms of text-to-image similarity.

---

> ### Author Response · Authors · 2021-11-20
> **Part 2: Response to Reviewer rpKE**
>
> >In Section 3.2.1 “Extreme Modulating Tensor Factorization”, what are the outputs of the hyper network? Are they 4*R tensors (
>  )? What is R set to be? How is it ensured that the outputs have their corresponding ranks during the course of training? Please carify these details.
>
> A modulating tensor can be represented by a sum of R outer products of rank-1 tensors; see [R1]. Here, R denotes the rank of the tensor. Depending on a type of embeddings,
>
> - Sentence embedding: hypernetworks generate only 4 rank-1 tensors to build a modulating tensor of rank R=1 for modulating convolutional layers in StyleGAN2. That is, according to Eq 3, R=1 and modulating tensor T simply becomes $T = t^r_{out} \otimes t^r_{in} \otimes t^r_h \otimes t^r_w$, where $t^r_{out}$,  $t^r_{in}$ ,  $t^r_{h}$, $t^r_{w}$  are rank-1 tensors with dimensions $out$,$in$, $h$ and $w$ respectively, produced by hypernetworks. We experimented to generate modulating tensors of higher rank (R=5, R=10). However, they have not improved the results but increased the number of parameters in the model. In the case of continuous image generation using INR-based HyperCGAN, hypernetworks generate 2 rank-1 tensors to build a modulating matrix for linear layers in the main backbone. In this setup, we set R to 5, so that the modulating matrix has a rank of 5.  In this case a modulating tensor is a sum of outer product of 5  $t^r_{out}$ and  $t^r_{in}$: $T = \sum_{r=1}^{5}t^r_{out} \otimes t^r_{in}$, as a sum of outer products of  r-th  $t^r_{out}$ and $t^r_{in}$ rank-1 tensors. We note that R=1 was not stable during training, and R=10 had similar performance as R=5, but the training process was slower.
>
> - Word Embeddings: hypernetworks generate 3 rank-1 tensors ($t^r_{in}$ ,  $t^r_{h}$, $t^r_{w}$) for each word, and later their outer product is used to obtain a word tensor of dimension 3. For each word, we build $T = t_{in} \otimes t_h \otimes t_w$, where  $t_{in}$ ,  $t_{h}$, $t_{w}$ rank-1 tensors produced by hypernetworks. This tensor will be reshaped and into a matrix, and a scaled dot product attention mechanism (Eq 5) is applied to attend to the relevant words to obtain modulating tensor.
>
> >How is it ensured that the outputs have their corresponding ranks during the course of training?
>
> Since hypernetworks in our method produce only rank-1 tensors (vectors), their rank is ensured during the course of training.
>
> [R1] (Henk A. L. Kiers, et.al, 2000) https://doi.org/10.1002/1099-128X(200005/06)14:3<105::AID-CEM582>3.0.CO;2-I
>
> >“To condition the final projection head in the discriminator on TDs(c), we use s = h >F(c), where h is the output of the last discriminator hyper-block, s is the output of the discriminator, and F(c), the tensor produced by our hyper-block.” Why is this done only for the sentence-level modulations?. Please also add dimensionalities to the corresponding variables.
>
> Will be answered in Part 4
>
> >Section 3.2.2 "Word-Level Modulation" can be motivated better. For, eg it seems like it is a cross-attention layer between the convolutional weights and the sequence. I suggest to run an ablation, removing this layer to showcase its importance.
>
> Thank you for the suggestion.  Removing the word-level modulation layer without adding other conditioning mechanisms, will reduce our models to unconditional GANs. However, our additional ablations replacing word-level modulation with sentence-level modulation, shows the effectiveness of our word-level modulation as shown in Table 2 in the paper.
>
> >INR GAN has both a generator network and an MLP. Is the proposed modulation applied only to the MLP? Why?
>
> In INR-GAN, there are two modules:
> 1.Hypernetwork which takes a noise vector z and generates parameters for the generator network.
> 2.A generator network (MLP model) is constructed from the output of a former hypernetwork along with a shared weight as detailed in the INR-GAN paper to independently predict RGB pixel values for each location in an input coordinate grid.
>
> To facilitate conditioning on the input text, we modulate the parameters of the generator network (MLP in INR-GAN case) with either the proposed sentence-level or word-level mechanisms.
>
> >What architecture is the Text Encoder? How is the output pooled from shape (sequence length, hidden size) to (hidden size,) to form the sentence embedding? Is this trained end-to-end?
>
> We adopt pretrained text encoder from AttnGAN. This text encoder is used in all the baselines reported in the paper. Therefore, for consistency, we also used AttnGAN text encoder which is based on a bi-directional Long Short-Term Memory (LSTM). In the bi-directional LSTM, each word corresponds to two hidden states, one for each direction. To represent the semantic meaning of a word,  they concatenate its two hidden states. The last hidden states of the bi-directional LSTM are concatenated to be the global sentence vector. The hidden size of both embeddings is equal to 256. We included it in the Appendix 6.4.

---

> ### Author Response · Authors · 2021-11-20
> **Part 3: Response to Reviewer rpKE**
>
> >In the human evaluations are the images from DF-GAN, DM-GAN and HyperCGAN shown at the same time or one after the other. In either case, what is the amount of time the human rater gets between two sets of images? Please add these details to make the study reproducible. Also, consider adding them to Table 1.
>
> Yes, during human evaluation generations from baselines and our model are shown at the same time. The amount of time the users spent to choose the image was at least 15 seconds. We specifically asked users to look carefully at generations and pick the closest one to the text description. Generations were shown at random order for each task.  Results are summarized in Table 3, 4, 5 in the updated paper.
>
> >Other details
>
> >It will be nice if the DAMSM loss is described in the paper to make it more self contained.
>
> We added DAMSM loss details as appendix 6.5 in the updated paper paper.
>
>
> >“When using sentence-level conditioning, we use an eightlayer MLP for the shared backbone that we call TGs body and a single-layer MLP as the output layer (TGs head), which is different for each generator block. For word-level conditioning, we use separate Conv 1x1 layers for each block in the generator (TGw block); see the Fig. 2 generator part”: A single-layer MLP and a Conv 1x1 for each block in the generator is equivalent?
>
> No, in order to share weights across all word embeddings, we used conv1x1 over words. We also want to highlight that, "eight-layer" is a typo in the paper. We used 2-layer MLP., that we fixed that in the updated version,  reflected in Figure 2.
>
> >Consider summarizing the “extrapolation meaningfulness results in a new table.
>
> Thanks for the suggestion, the results are summarized in Table 4, 5.
>
> >Typos
>
> >The output of the backbone is consumed by the output layers, which then condition the generator via tensor modulation. What does “output layers” refer to?
>
> We meant “T_Gs head layers” and we clarified that in the updated version.
>
> >For all human studies, 250 generations were selected at random. For each image we assigned 5 participants in Amazon Mechanical Turk and collected 1500 responses in total. I suppose this should be 1250?
>
> We fixed it. Thank you!
>
> > I recommend that the authors cite other (non-GAN based) related text-to-image synthesis works.
> Thanks for sharing.
>
> We included some references in the updated version

---

> ### Author Response · Authors · 2021-11-20
> **Part 4: Response to Reviewer rpKE**
>
> “To condition the final projection head in the discriminator on TDs(c), we use s = hT * F(c), where h is the output of the last discriminator hyper-block, s is the output of the discriminator, and F(c), the tensor produced by our hyper-block.” Why is this done only for the sentence-level modulations?. Please also add dimensionalities to the corresponding variables.
>
> Initially, it was a mismatch in the input dimension for F(c). For example, word embeddings had a size of (hidden size, sequence length).  We found two ways to condition the final projection head in D and experimented with HyperCGAN_StyleGAN2:
> 1.We average word embeddings across sequence length which will become of size (hidden size) and pass it through F(c).
> 2.We pass each word embeddings in the sequence through F(c) and then take the average of the output of F(c):
>
> |                            | DAMSM-R | CLIP-R | FID   |
> |----------------------------|---------|--------|-------|
> | Average of word embeddings | 47.58%  | 46.44% | 21.46 |
> | Average of output of F(c)  | 45.94%  | 43.77% | 25.86 |
>
> Results in the table above indicate that having F(c) for word conditioning did not improve the performance of HyperCGAN_StyleGAN2. Taking the average of word embeddings as an input to F(c) was comparable to the Discriminator without F(c). However, in the second case, the results were worsened both in terms of retrieval scores and image quality. e.g. CLIP-R score was 43.77% and the FID score increased to 25.86. As for dimensionality, h has a dimensionality of 512, word embeddings have 256, F(c) takes 256-dimensional input and produces 512-dimensional output.

---

> > ### Comment · Reviewer_rpKE · 2021-11-24
> > **Followup Questions (2)**
> >
> > Thanks for the additional ablations. Just to confirm, these ablations are trained without DAMSM loss?

---

> > > ### Author Response · Authors · 2021-11-29
> > > **Re: Followup Questions (2)**
> > >
> > > Yes, these ablations are trained without DAMSM loss.

---

> ### Author Response · Authors · 2021-11-22
> **Additional experiments on CUB dataset**
>
> Additionally, we trained our models on a 256x256 CUB dataset and compared to the recent backbones (DM-GAN, DF-GAN).
>
> | Model               |                   | DAMSM-R | CLIP-R |         FID |
> |---------------------|-------------------|:-------:|:------:|------------:|
> | DM-GAN              |                   |  75.89  |  11.98 |       16.09 |
> | DF-GAN              |                   |  39.05  |  10.29 |       14.81 |
> |                     |                   |         |        |             |
> | HyperCGAN_StyleGAN2 | sent              |  25.51% | 15.82% | 13.29 |
> | HyperCGAN_StyleGAN2 | sent + DAMSM loss |  28.53% | 19.07% | 11.72 |
> | HyperCGAN_StyleGAN2 | word              |  25.55% | 19.94% |  11.80 |
> | HyperCGAN_StyleGAN2 | word + DAMSM loss |  68.53% | 21.45% | 15.02 |
> | HyperCGAN_INR       | sent              |  18.43% | 17.85% | 20.82 |
> | HyperCGAN_INR       | sent + DAMSM loss |  44.15% | 26.94% |    25.54    |
> | HyperCGAN_INR       | word              |  17.46% | 17.95% | 18.73 |
> | HyperCGAN_INR       | word + DAMSM loss |  46.39% | 31.90% | 15.72 |
> | Real                |                   |  18.24% | 25.47% |             |
>
> We performed additional experiments where we trained our models on CUB dataset and compared them to recent baselines (DF-GAN, DM-GAN). In Table 7, the results indicate that our models achieve high CLIP-R precision and comparable FID score compared to baselines.

---

### Official Review · Reviewer_yoBb · 2021-10-28

**Correctness:** 3
**Technical Novelty And Significance:** 2
**Empirical Novelty And Significance:** 2
**Recommendation:** 5
**Confidence:** 4

**Main Review:**

Pros:
1. It is first paper of using hypernetworks for text-to-image generation. It looks interesting.

2. Utilizing hypernetworks  inevitably suffer from  memory-intensive problem, since it synthesizes the weights, especially for super network (e.g., StyleGAN), authors contribute some extend energy to fix it, which is convincing.

3. Different architectures and datasets are explored to validate the generalization.

Cons:
1. I fail to get the intuition why we need hypernetworks for text-to-image. The paper could not show clearly the advantage when using hypernetworks. Compared to condition the text description on batchnorm layer, hypernetworks leads to more learned weights.

2. Using hypernetworks has weak contribution in community, since we could insert hypernetworks into any current existed architecture.

3. The reported results are not convincing, since both the baselines and the proposed method have different architecture. In fact, the proposed method benefits from the well-devised stylegan architecture. I am wondering what is the performance if we use stylegan-based architecture for text/audio-to-image translation.   Could authors show the number of parameters  (MB) and GFLOPs?

4. It seems the proposed method has little advantage based on the reported results, which indicate that hypernetworks does not work well. In fact, what is the result if author remove the hypernetworks form the discriminator, and directly leverage the same one of styleganv2. I believe it improves the performance.

5.  There are a few errors  in Figure 2, such as the overlap arrows and the repeated TGs in caption.  I do not appreciate the figure2, which is not beautiful and enjoyable. I would like authors to improve it.


-------------After rebuttal------------------

Thanks for authors' response.  I still feel negative about this paper. One reason is that the proposed method is simple, which seems to use HyperNetworks for text-to-image generation. I fail to get the convincing intuition why Hypernetworks is perfect for this task. Second one is the advantage of the proposed method benefits from the stylegan-based architecture ($\mathbf{Reviewer rpKE }$ also mentions it). To be summary, I am negative to this paper.

**Summary Of The Paper:**

Authors explore the application of hypernetworks for text-to-image generation. Given the input text description, the hypernetworks learn the weights of convolution. Instead of directly leveraging hypernetworks, author propose to modulate the convolutional weights  by the generated weights. To validate the generalization of the proposed method, authors perform experiments on two generation architecture: StyleGAN and INR-GAN. Furthermore,  since hypernetworks lead to memory-intensive problem when using it to generate the weights  for each block, authors introduce new method to address this issue. The quantitative and qualitative results support the effectiveness of the proposed method.

**Summary Of The Review:**

Although this paper first explore text-to-image generation by using hypernetwork, I am wondering the intuition and the contribution. If only using hypernetworks, I think it is limited to be accepted.

---

> ### Author Response · Authors · 2021-11-20
> **Part 1: Response to Reviewer yoBb**
>
> Thank you for your detailed review, questions and suggestions. We here address them and incorporate the feedback.
>
> >I fail to get the intuition why we need hypernetworks for text-to-image. The paper could not show clearly the advantage when using hypernetworks. Compared to condition the text description on batchnorm layer, hypernetworks leads to more learned weights.
> Using hypernetworks has weak contribution in community, since we could insert hypernetworks into any current existed architecture.
>
> There are two types of hyper-networks in our system: (a) text conditioning hypernetwork that we formulate as modulation mechanisms (sentence or word-level), (b) generation hypernetwork that we adopt for continuous image generation. We demonstrated that HyperCGAN is a flexible framework that is capable of generating continuous images based on unconditional INR-GANs generation HyperNet as well as discrete images based on StyleGAN2 . The continuous image generation ability, facilitated by hypernetwork modulation and generation,  gives the model a natural ability  to perform text-to-super-resolution-image out of the box, and extrapolate outside the training boundaries as we demonstrated in our experiments; see Tables 3,4, and 5 and Fig 1 and 5.

---

> ### Author Response · Authors · 2021-11-20
> **Part 2: Response to Reviewer yoBb**
>
> >The reported results are not convincing, since both the baselines and the proposed method have different architecture. In fact, the proposed method benefits from the well-devised stylegan architecture. I am wondering what is the performance if we use stylegan-based architecture for text/audio-to-image translation. Could authors show the number of parameters (MB) and GFLOPs?
>
> Here we report the result of standard conditioning using our StyleGAN2 and INR-GAN backbones without our method. While we observe that the image quality of these baselines is a little better in terms of FID score, they significantly underperform in R-prec metrics indicating the lack of relevance of the generated image to the input text. This shows that standard conditioning does not work well on these architectures and proper design of the conditioning learning signal is necessary to achieve good performance. The table below also shows that existing backbones benefit from our proposed word level self-attention modulation (Eq 5 in the paper), which is at the heart of our paper.
>
> | Model         | sent | word | damsm loss | R      | CLIP-R | FID     |
> |---------------|------|------|------------|--------|--------|---------|
> | ** StyleGAN2 **     | x    |      |            | 47.16% |  44.13 | 17.1795 |
> | HyperCGAN_SG2 | x    |      |            | 50.11% |  49.85 |   23.59 |
> | HyperCGAN_SG2 | x    |      | x          | 64.04% |  54.45 |   31.47 |
> | HyperCGAN_SG2 |      | x    |            | 48.49% |  47.41 |   20.64 |
> | HyperCGAN_SG2 |      | x    | x          | 67.92% |  61.49 |   20.81 |
> |**  INR Base  **       | x    |      |            | 38.61% |  34.91 |   27.73 |
> | HyperCGAN_IR  | x    |      |            |  40.36 |  40.81 |   28.29 |
> | HyperCGAN_INR | x    |      | x          |  39.92 |   52.3 |   38.66 |
> | HyperCGAN_INR |      | x    |            |  35.67 |  37.23 |   25.39 |
> | HyperCGAN_INR |      | x    | x          |  56.98 |  52.64 |   30.44 |
>
> Additionally, we report the number of parameters for the baselines and our models. Apart from DFGAN,  AttnGAN, ControlGAN and DMGAN baselines have 3 discriminators, adopting a multistage generation pipeline. However, our method employs only a single modulated discriminator, adopting a single stage generation pipeline similar to DFGAN.  Here, we summarize the model parameters for each of the methods (baselines) in our paper:
>
> | Model                           | G          | D1         | D2         | D3          | Total number of parameters in D | FID     |
> |---------------------------------|------------|------------|------------|-------------|---------------------------------|---------|
> | AttnGAN                         | 13,811,456 | 13,304,450 | 42,800,258 | 160,774,274 | 216,878,982                     | 17.1795 |
> | ControlGAN                      | 36,733,888 | 13,304,450 | 42,800,258 | 160,774,274 | 216,878,982                     |   23.59 |
> | DM-GAN                          | 22,335,314 | 1,878,242  | 5,155,810  | 18,264,546  | 25,298,598                      |   31.47 |
> | DF-GAN                          | 12,240,010 | 20,170,566 |            |             | 20,170,566                      |   20.64 |
> | HyperCGAN StyleGAN2 (word)      | 29,661,154 | 26,331,649 |            |             | 26,331,649                      |   20.81 |
> | HyperCGAN  StyleGAN2 (sentence) | 25,923,504 | 25,949,156 |            |             | 25,949,156                      |   27.73 |
> | HyperCGAN_IR                    | x          |            |            |       40.36 |                           40.81 |   28.29 |
> | HyperCGAN_INR                   | x          |            | x          |       39.92 |                            52.3 |   38.66 |
> | HyperCGAN_INR                   |            | x          |            |       35.67 |                           37.23 |   25.39 |
> | HyperCGAN_INR                   |            | x          | x          |       56.98 |                           52.64 |   30.44 |
>
> [1] Karras, Tero, et al. "Analyzing and improving the image quality of stylegan." Proceedings of the IEEE/CVF Conference on Computer Vision and Pattern Recognition. 2020.

---

> ### Author Response · Authors · 2021-11-20
> **Part 3: Response to Reviewer yoBb**
>
> >It seems the proposed method has little advantage based on the reported results, which indicate that hypernetworks does not work well. In fact, what is the result if author remove the hypernetworks form the discriminator, and directly leverage the same one of styleganv2. I believe it improves the performance.
>
> Thanks for the suggestion. We replaced our modulated Discriminator to StyleGAN2’s one. We observe that our Modulated Discriminator helps to improve Retrieval scores (DAMSM-R and CLIP-R) with a little decrease in FID. This shows the importance of сonditioning Discriminator by our modulation technique to increase the relevance of the generated image to the input text. Table below summarizes the results:
>
> | Discriminator Conditioning        | DAMSM-R | CLIP-R |  FID  |
> |-----------------------------------|:-------:|:------:|:-----:|
> | HyperCGAN_StyleGAN2 + StyleGAN2 D | 44.19   | 41.13  | 21.37 |
> | HyperCGAN_StyleGAN2 + Modulated D | 50.11   | 49.85  | 23.59 |
>
> >There are a few errors in Figure 2, such as the overlap arrows and the repeated TGs in caption. I do not appreciate the figure2, which is not beautiful and enjoyable. I would like authors to improve it.
>
> Thank you for your suggestion. We replaced Figure 2 with a new version.

---

> ### Author Response · Authors · 2021-11-22
> **Additional Experiments on CUB dataset**
>
> Additionally, we trained our models on a 256x256 CUB dataset and compared to the recent backbones (DM-GAN, DF-GAN).
>
> | Model               |                   | DAMSM-R | CLIP-R |         FID |
> |---------------------|-------------------|:-------:|:------:|------------:|
> | DM-GAN              |                   |  75.89  |  11.98 |       16.09 |
> | DF-GAN              |                   |  39.05  |  10.29 |       14.81 |
> |                     |                   |         |        |             |
> | HyperCGAN_StyleGAN2 | sent              |  25.51% | 15.82% | 13.29 |
> | HyperCGAN_StyleGAN2 | sent + DAMSM loss |  28.53% | 19.07% | 11.72 |
> | HyperCGAN_StyleGAN2 | word              |  25.55% | 19.94% |  11.80 |
> | HyperCGAN_StyleGAN2 | word + DAMSM loss |  68.53% | 21.45% | 15.02 |
> | HyperCGAN_INR       | sent              |  18.43% | 17.85% | 20.82 |
> | HyperCGAN_INR       | sent + DAMSM loss |  44.15% | 26.94% |    25.54    |
> | HyperCGAN_INR       | word              |  17.46% | 17.95% | 18.73 |
> | HyperCGAN_INR       | word + DAMSM loss |  46.39% | 31.90% | 15.72 |
> | Real                |                   |  18.24% | 25.47% |             |
>
> We performed additional experiments where we trained our models on CUB dataset and compared them to recent baselines (DF-GAN, DM-GAN). In Table 7, the results indicate that our models achieve high CLIP-R precision and comparable FID score compared to baselines.

---

### Official Review · Reviewer_Jcwi · 2021-11-02

**Correctness:** 3
**Technical Novelty And Significance:** 3
**Empirical Novelty And Significance:** 2
**Recommendation:** 6
**Confidence:** 3

**Main Review:**

The paper proposes to train a Hypernet which directly modulates the weights of an underlying generator to generate images based on a condition. The training itself is done with the same losses as traditional GANs.

One of the novelties is how to structure the Hypernet to make it more efficient. Naively modulating the weights would imply to predict a modulation parameter for each parameter in the GAN which is inefficient and may not be computationally possible. Instead, the authors propose to use a low-rank decomposition to make the approach computationally feasible. They also show how to apply word-level modulation through their Hypernet.

The experiments evaluate their approach on both StyleGAN2 and INR-GAN on the COCO and ArtEmis datasets. The results are evaluated quantitatively based on IS/FID (for image quality) and R-precision/CLIP-R for text-image alignment. Additionally, user studies for text-image alignment were performed. The new approach outperforms current baselines (e.g. DM-GAN and DF-GAN) based on FID and CLIP-R which are arguably the two most important metrics.

My main question is about the exact motivation of this approach. You state that this is more powerful than the original approaches of directly conditioning the model on the text. It would be useful to have concrete examples of how/when your model is better or at least motivate in which settings your model would be (theoretically) better than current approaches. One of the motivations (from my point of view) may be that you can apply this approach directly to GANs that can produce high-res images like StyleGAN2. Have you tried using this to generate higher-res images (>256px)?

That said, how do you obtain the underlying GANs? Do you use pre-trained models (if yes, pre-trained on what)? Or do you use randomly initialized models?
Also, have you looked at other evaluation approaches for text-image alignment, e.g. SOA [1]?
Could your approach be used for more fine-grained control over also object arrangement (can be evaluated by SOA) and relations between objects?
Also, XMCGAN seems to be missing from the baselines, both qualitative and quantitative.


[1] Semantic Object Accuracy for Generative Text-to-Image Synthesis, TPAMI, 2020

**Summary Of The Paper:**

The paper proposes a novel approach for text-to-image synthesis. Instead of training the weights of a GAN directly the approach is to train a HyperNet which modulates the GAN's weights based on the text condition. The approach is evaluated for both StyleGAN2 and INR-GAN (a continuous GAN) and obtains good results compared to current baselines.

**Summary Of The Review:**

The paper proposes a new way of text-to-image generation by directly modulating GAN weights conditioned on the text input. The results seem promising and the evaluation is extensive. It would be interesting if this approach can scale to higher resolutions and can generate more complex images where we can also condition on invidiual object locations etc.

---

> ### Author Response · Authors · 2021-11-20
> **Part 1: Response to Reviewer Jcwi**
>
> Thank you for your detailed review, questions, and suggestions.
>
> >how do you obtain the underlying GANs? Do you use pre-trained models (if yes, pre-trained on what)? Or do you use randomly initialized models?
>
> The parameters of underlying GANs (StyleGAN2 and INR-GAN) are initialized randomly. We condition them using our method and train from scratch.
>
> We are currently performing some experiments and will respond within 24 hours to the remaining questions in Part 2.

---

> > ### Author Response · Authors · 2021-11-22
> > **Part 2: Response to Reviewer Jcwi**
> >
> > >My main question is about the exact motivation of this approach. You state that this is more powerful than the original approaches of directly conditioning the model on the text. It would be useful to have concrete examples of how/when your model is better or at least motivate in which settings your model would be (theoretically) better than current approaches. One of the motivations (from my point of view) may be that you can apply this approach directly to GANs that can produce high-res images like StyleGAN2. Have you tried using this to generate higher-res images (>256px)?
> >
> > We appreciate Reviewer Jcwi’s valuable suggestion. We demonstrated that HyperCGAN is a flexible framework that is capable of generating continuous images based on unconditional INR-GANs generation HyperNet as well as discrete images based on StyleGAN2 . The continuous image generation ability, facilitated by hypernetwork modulation and generation,  gives the model a natural ability to extrapolate outside the training boundaries as we demonstrated in our experiments; see Tables 3,4, and 5 and Fig 1 and 5. We focused on 256x256 for fair comparisons with SotA methods, and we are not aware of existing benchmarks on text-to-high-resolution images. We applied our approach to condition the StyleGAN2 backbone using word-level modulation to generate high-resolution images. Specifically, we tried to generate 1024 x 1024 images from COCO captions. In terms of retrieval scores DAMSM-R and CLIP-R, we obtained 47.84% and 42.07%, respectively. However, in terms of FID, the result is 36.77. We also attach some qualitative results for that in Appendix 6.7.
> >
> > | Model                                                | DAMSM-R | CLIP-R |  FID  |
> > |------------------------------------------------------|:-------:|:------:|:-----:|
> > | HyperCGAN_StyleGAN2 1024x1024 (trained from scratch) | 47.84%  | 42.07% | 36.77 |
> >
> > >Also, have you looked at other evaluation approaches for text-image alignment, e.g. SOA [1]? Could your approach be used for more fine-grained control over also object arrangement (can be evaluated by SOA) and relations between objects?
> >
> > Thanks so much for the suggestion. We agree that SOA could be suitable for the COCO dataset but not for ArtEmis dataset due to its artistic nature and the existence of abstract art in its categories. Hence, we focused on FID/CLIP-R/DAMSM-R as well as human evaluation to measure the performance. We worked on SOA for COCO, but we realized that it is computationally expensive, around 400,000 images need to be generated for each dataset, and then running object detection on top of them is taking a lot of time. We aim to get SOA numbers for COCO and report them in the final version.
> >
> > >Also, XMCGAN seems to be missing from the baselines, both qualitative and quantitative.
> >
> > As we report in Section 2 of our paper,  we were unable to reproduce the FID scores of XMC-GAN using their repository.  While the XMC-GAN authors have released their JAX code [1] , they have neither released their model nor generated images and did not respond to our emails asking for them. Preliminary training runs on our V100 instances produced far worse FID scores than those reported in the paper. To reproduce their full results would require a 32-core Pod slice of Google Cloud TPU v3 devices. The codebase also does not include an implementation of their proposed metric or an inference script to generate images.
> >
> > [1] https://github.com/google-research/xmcgan_image_generation

---

> ### Author Response · Authors · 2021-11-22
> **Additional Experiments on CUB dataset**
>
> Additionally, we trained our models on a 256x256 CUB dataset and compared to the recent backbones (DM-GAN, DF-GAN).
>
> | Model               |                   | DAMSM-R | CLIP-R |         FID |
> |---------------------|-------------------|:-------:|:------:|------------:|
> | DM-GAN              |                   |  75.89  |  11.98 |       16.09 |
> | DF-GAN              |                   |  39.05  |  10.29 |       14.81 |
> |                     |                   |         |        |             |
> | HyperCGAN_StyleGAN2 | sent              |  25.51% | 15.82% | 13.29 |
> | HyperCGAN_StyleGAN2 | sent + DAMSM loss |  28.53% | 19.07% | 11.72 |
> | HyperCGAN_StyleGAN2 | word              |  25.55% | 19.94% |  11.80 |
> | HyperCGAN_StyleGAN2 | word + DAMSM loss |  68.53% | 21.45% | 15.02 |
> | HyperCGAN_INR       | sent              |  18.43% | 17.85% | 20.82 |
> | HyperCGAN_INR       | sent + DAMSM loss |  44.15% | 26.94% |    25.54    |
> | HyperCGAN_INR       | word              |  17.46% | 17.95% | 18.73 |
> | HyperCGAN_INR       | word + DAMSM loss |  46.39% | 31.90% | 15.72 |
> | Real                |                   |  18.24% | 25.47% |             |
>
> We performed additional experiments where we trained our models on CUB dataset and compared them to recent baselines (DF-GAN, DM-GAN). In Table 7, the results indicate that our models achieve high CLIP-R precision and comparable FID score compared to baselines.

---

### Official Review · Reviewer_p56q · 2021-11-03

**Correctness:** 3
**Technical Novelty And Significance:** 2
**Empirical Novelty And Significance:** 2
**Recommendation:** 5
**Confidence:** 4

**Main Review:**

Strengths:
1. The implementation of the hyper-networks to control the weights in both generator and discriminator.
2. A new dataset ArtEmis is adopt to evaluate the performance.
3. The continuous image generation is explored.

Weaknesses:
1. The main technical contribution is the implementation of the hyper-networks in text-to-image generation tasks, I would expect to have more discussion about this implementation, like why text-conditioned hyper-network can improve the text-to-image generation task.

2. In Table 2, the method based on StyleGAN2 with word conditioning and DAMSM loss is the selected best method proposed by authors, shown in Table 1. I am confused after the method also adopts the DAMSM loss, similar as other AttnGAN etc., the DAMSM-R is still not good, contracted to the claim that AttnGAN etc., achieve better DAMSM-R due to the adoption of DAMSM loss. Similar confusion about HyperCGAN INR with word conditioning and DAMSM loss.

3. I am confused that the CLIP-R score for real images is much higher than the score for DAMSM-R. As I know, CLIP also adopts similarity calculation to train the model, why there is such significant difference between DAMSM-R and CLIP-R. Also, it would be better if authors can provide more details about how to calculate these two scores on real images.

4. Although authors mentioned their method can be applied to continuous image generation, there are not many discussions and experiments about this shown in the paper.

**Summary Of The Paper:**

The paper adopts hyper-networks in text-to-image generation, which is used to control weights in both generator and discriminator, based on the provided text descriptions. Then, authors evaluate their method in traditional image generation and continuous image generation.

**Summary Of The Review:**

The paper evaluates the implementation of hyper-networks on text-to-image generation, for the technical novelty, it is not quite significant. Also, The discussion about the continuous image generation mentioned in the paper is not sufficient.

---

> ### Author Response · Authors · 2021-11-20
> **Part 1: Response to Reviewer  p56q**
>
> We thank Reviewer p56q for the valuable input. We address below the raised concerns.
>
> >The main technical contribution is the implementation of the hyper-networks in text-to-image generation tasks, I would expect to have more discussion about this implementation, like why text-conditioned hyper-network can improve the text-to-image generation task.
>
> We believe that hypernet-based conditioning is a more expressive mechanism than the embedding-based conditioning approaches prevalent in existing T2I literature [1]. Text embeddings (sentence, word) compared to class labels are complex, since they contain complex relationships between words and their semantic representations. If one word is changed within the sentence, we would end up with a new embedding which can have a different representation. To handle this type of conditioning signal, we need different types of conditioning mechanisms that allow us to “adapt” a model for different conditioning inputs in an expressive way. Hypernetworks have this ability by producing weights based on the conditioning input. We demonstrated that HyperCGAN is a flexible framework that is capable of generating continuous images based on unconditional INR-GANs generation HyperNet as well as discrete images based on StyleGAN2.  The continuous image generation ability, facilitated by hypernetwork modulation and generation,  gives the model a natural ability  to perform text-to-super-resolution-image out of the box, and extrapolate outside the training image boundaries.
>
> [1] Tomer Galanti and Lior Wolf. On the modularity of hypernetworks. Advances in Neural Information Processing Systems, 33, 2020.
>
> >In Table 2, the method based on StyleGAN2 with word conditioning and DAMSM loss is the selected best method proposed by authors, shown in Table 1. I am confused after the method also adopts the DAMSM loss, similar as other AttnGAN etc., the DAMSM-R is still not good, contracted to the claim that AttnGAN etc., achieve better DAMSM-R due to the adoption of DAMSM loss. Similar confusion about HyperCGAN INR with word conditioning and DAMSM loss.
>
> The similarity score of DAMSM-R is calculated using Deep Attentional Multimodal Similarity Model (DAMSM) which maps each subregion of an image and its corresponding word in the sentence to a joint embedding space. On the other hand, CLIP adopts a contrastive objective to learn to bring visual and textual embeddings together. We follow the R1-precision calculation used in previous works which utilizes DAMSM to compute similarity. In recent literature (e.g.,[1,2]) it has been already established that the DAMSM-R score is not a reliable score for text2image generation. [2] found that models which are not optimized for the DAMSM module, underperform in the DAMSM-R. To achieve a high score in this metric, models need to include the DAMSM-module as a part of their training process. [2] finds the DAMSM-R metric to be model-specific metric and suggests the adoption of a CLIP-based evaluation metric. In Table 1, the results on real images show that CLIP-based R-prec achieves a significant high-retrieval score compared to DAMSM-based R-prec on COCO. The performance of real images on ArtEmis also suggests that CLIP should be adopted in our study to compute R-prec.
>
> >I am confused that the CLIP-R score for real images is much higher than the score for DAMSM-R. As I know, CLIP also adopts similarity calculation to train the model, why there is such significant difference between DAMSM-R and CLIP-R. Also, it would be better if authors can provide more details about how to calculate these two scores on real images.
>
> The reason behind the low score in DAMSM-R even though we adopt DAMSM loss might be that we use DAMSM partially. DAMSM loss is a sum of two losses defined as $L_{DAMSM} = L^{w1} + L^{w2} + L^{s1} + L^{s2}$ where $L^{s1}, L^{s2}$ are referred as a sentence loss and $L^{s1}, L^{s2}$ are referred as a word loss. More detailed information of DAMSM loss is included in Appendix 6.5. Since we either adopt sentence embeddings or word embeddings we do not use full DAMSM but rather the sentence level loss or the word level loss depending on the input embeddings. For example, when sentence embeddings are used we utilize $L^{s1} + L^{s2}$, and when word embeddings are used as a condition $ L^{w1} + L^{w2}$. Even though we achieved a lower score in DAMSM-R, in previous answer we showed that is not a good metric for calculating R-prec.
>
> >Although authors mentioned their method can be applied to continuous image generation, there are not many discussions and experiments about this shown in the paper.
>
> Will be in Part 2

---

> > ### Author Response · Authors · 2021-11-22
> > **Part 2: Response to Reviewer p56q**
> >
> > >Although the authors mentioned their method can be applied to continuous image generation, there are not many discussions and experiments about this shown in the paper.
> >
> > As a gentle notice, we mentioned in the paper that our word-level modulation aims at grounding words to independent pixels at coordinates (x, y) in an input grid which is represented as low-res features at earlier layers and the final RGB value in the last layer. In Figure 1 and Figure 5, we showed the property of HyperCGAN_INR model to extrapolate beyond training resolutions. Human studies in Tables 4, 5 indicate that our HyperCGAN_INR extrapolations are meaningful and do not make generated images less aligned with a text description.
> >
> > We applied our method to the INR-GAN backbone. In addition to the extrapolation property, our HyperCGAN_INR model can generate higher-resolution images through superresolution even though the model was trained on low-resolution images like 256 x 256. As a contrast to the superresolution property of HyperCGAN-INR, we trained HyperCGAN_StyleGAN2 to generate 1024x1024 images from COCO captions from scratch. Below we report the results for high-resolution text-to-image generations for two models for comparison:
> >
> >
> > | Model                                                                                   | DAMSM-R |  CLIP-R |  FID  |
> > |-----------------------------------------------------------------------------------------|:-------:|:-------:|:-----:|
> > | HyperCGAN_StyleGAN2 1024x1024 (trained from scratch)                                    | 47.84%  | 42.07%  | 36.77 |
> > | HyperCGAN_INR-GAN 1024x1024  (Out-of-the-box superresolution), model trained on 256x256 | 53.04 % | 39.79 % | 44.24 |
> >
> > HyperCGAN_StyleGAN2 achieves lower FID score 36.77 compared to HyperCGAN_INR-GAN with 44.24 FID score. Qualitatively, generations of HyperCGAN_StyleGAN2 are of higher quality than ones obtained through HyperCGAN_INR-GAN superresolution. The underlying INR-GAN backbone can also be trained for generating high-resolution images, but we could not train it in the limited time period because compared to the discrete backbone, the training process of HyperCGAN_INR is much slower. Nevertheless, we believe we may get comparable results to HyperCGAN_Stylegan2 if we train INR-GAN based version from scratch as well.

---

> ### Author Response · Authors · 2021-11-22
> **Additional experiments on CUB dataset**
>
> Additionally, we trained our models on a 256x256 CUB dataset and compared to the recent backbones (DM-GAN, DF-GAN).
>
> | Model               |                   | DAMSM-R | CLIP-R |         FID |
> |---------------------|-------------------|:-------:|:------:|------------:|
> | DM-GAN              |                   |  75.89  |  11.98 |       16.09 |
> | DF-GAN              |                   |  39.05  |  10.29 |       14.81 |
> |                     |                   |         |        |             |
> | HyperCGAN_StyleGAN2 | sent              |  25.51% | 15.82% | 13.29 |
> | HyperCGAN_StyleGAN2 | sent + DAMSM loss |  28.53% | 19.07% | 11.72 |
> | HyperCGAN_StyleGAN2 | word              |  25.55% | 19.94% |  11.80 |
> | HyperCGAN_StyleGAN2 | word + DAMSM loss |  68.53% | 21.45% | 15.02 |
> | HyperCGAN_INR       | sent              |  18.43% | 17.85% | 20.82 |
> | HyperCGAN_INR       | sent + DAMSM loss |  44.15% | 26.94% |    25.54    |
> | HyperCGAN_INR       | word              |  17.46% | 17.95% | 18.73 |
> | HyperCGAN_INR       | word + DAMSM loss |  46.39% | 31.90% | 15.72 |
> | Real                |                   |  18.24% | 25.47% |             |
>
> We performed additional experiments where we trained our models on CUB dataset and compared them to recent baselines (DF-GAN, DM-GAN). In Table 7, the results indicate that our models achieve high CLIP-R precision and comparable FID score compared to baselines.

---

### Author Response · Authors · 2021-11-20
**Paper Revised Version**

We thank the reviewers for their valuable feedback. We thank the reviewers for appreciating our exploration in text-to-continous image generation (Reviewer p56q, Reviewer yoBb, Reviewer rpKE), and our evaluation on ArtEmis dataset (Reviewer p56q, Reviewer yoBb). We are glad that they found our approach to be interesting and novel (Reviewer Jcwi and yoBb), and the extreme low-rank tensor decomposition technique convincing (Reviewer Jcwi, Reviewer yoBb). We are pleased that Reviewer rpKE recognizes that our work is the first to augment StyleGAN v2 for text-to-image synthesis. We address below the raised concerns and incorporate all feedback.

Key updates in the revised version are listed below; including additional results/improvements requested by the reviewers.

* Figure 1 and Figure 2 were improved and updated
* Table 2 now includes underlying backbones with standard conditioning as  a baseline
* Table 3,4,5 for Human Studies are added
* Per the reviewer’s  request, we added DAMSM loss details as Appendix 6.5
* Some typos are fixed and some texts are added as an attempt to increase clarity in the paper.
* Appendix was updated
* New results on CUB dataset is reported in Appendix 6.6

---

### Decision · Program_Chairs · 2022-01-20

**Decision:**

Reject

**Comment:**

After the author response multiple reviewers remained concerned over the degree to which the current manuscript makes the case for the proposed hyper-network approach to text-to-image generation. It was felt that this was mainly an empirical paper for which the reviewers remain unconvinced that the proposed hyper-network based modulation was better than simple channel-wise StyleGAN2 style modulation. While the authors have shown that their approach beats a StyleGAN2 baseline with sentence conditioning on CLIP-R the reviewers felt that the comparisons with StyleGAN2 baseline needed fairer word conditioning. Only one reviewer recommended accepting this paper.

The AC recommends rejection.